# On the Planning Abilities of Large Language Models : A Critical Investigation

**Karthik Valmeekam**
School of Computing & AI
Arizona State University Tempe.
`kvalmeek@asu.edu`

**Matthew Marquez**
School of Computing & AI
Arizona State University, Tempe.
`mmarqu22@asu.edu`

**Sarath Sreedharan**\*
Department of Computer Science,
Colorado State University, Fort Collins.
`sarath.sreedharan@colostate.edu`

**Subbarao Kambhampati**
School of Computing & AI
Arizona State University, Tempe.
`rao@asu.edu`

## Abstract

Intrigued by the claims of emergent reasoning capabilities in LLMs trained on general web corpora, in this paper, we set out to investigate their planning capabilities. We aim to evaluate (1) the effectiveness of LLMs in generating plans autonomously in commonsense planning tasks and (2) the potential of LLMs as a source of heuristic guidance for other agents (AI planners) in their planning tasks. We conduct a systematic study by generating a suite of instances on domains similar to the ones employed in the International Planning Competition and evaluate LLMs in two distinct modes: *autonomous* and *heuristic*. Our findings reveal that LLMs' ability to generate executable plans autonomously is rather limited, with the best model (GPT-4) having an average success rate of ~12% across the domains. However, the results in the heuristic mode show more promise. In the heuristic mode, we demonstrate that LLM-generated plans can improve the search process for underlying sound planners and additionally show that external verifiers can help provide feedback on the generated plans and back-prompt the LLM for better plan generation.

## 1 Introduction

It would be no exaggeration to say that transformer-based large language models (LLMs) have revolutionized the field of natural language processing (NLP). Kicked off by the advances presented by the GPT-x models developed by OpenAI [27], these types of language models currently provide state-of-the-art performance in many of the standard NLP tasks. Although LLMs were originally developed mostly to do word sequence completion tasks, with no guarantees about the completion beyond its coherence, there have been increasing claims and anecdotal evidence that they have other *emergent capabilities* that are not normally associated with sequence completion. Indeed, the hints of such emergent capabilities has started a veritable land rush, with researchers probing (prompting) and studying LLM behavior almost as if they were artificial organisms (c.f. [16]). Of particular interest to us in this paper is the thread of efforts that aim to investigate (and showcase) reasoning abilities of LLMs–including commonsense reasoning [35, 29, 7], logical reasoning [33], and even ethical reasoning [15]. The macro-tenor of the drumbeat of these works has been suggesting that LLM's are indeed capable of doing such kinds of reasoning [19, 37, 4].

---

\*Author was at Arizona State University during part of this work

37th Conference on Neural Information Processing Systems (NeurIPS 2023).

One type of reasoning task that has been well studied in the AI community is planning and sequential decision making. At its simplest, planning involves developing a course of actions (policy) which when executed takes the agent to a desired state of the world. Planning has generally been studied primarily as an inference on world and reward models–whether specified by humans or learned by the agent by interacting with its world. In this paper, we are interested in seeing what planning abilities, if any, LLMs may already have, given their high capacity functions (with billions of tunable parameters) trained on web-scale corpora. Specifically, we are interested in answering two broad questions:

1. How effective are LLMs by themselves in generating simple plans in commonsense planning tasks (of the type that humans are generally quite good at)?

2. How good are LLMs in being a source of heuristic guidance for other agents in their planning tasks?

Notice that in theory, it is possible for LLMs to be very effective as idea generators for external sound planners or humans in the loop in computer-supported cooperative work scenarios, while themselves being very bad at generating plans that are guaranteed to be correct. This is especially likely because the chief power of LLMs comes from their pattern-finding abilities than from first-principles simulations over world models. Compared to a planner that is guaranteed to be correct in a narrow set of domains, LLMs may likely be good at generating plausible (but not guaranteed to be correct) plan heuristics/suggestions in many more domains.

To investigate these questions in a systematic rather than anecdotal manner, we generate a suite of planning problem instances [2] based on the kinds of domains employed in the International Planning Competition [14]. To eliminate the subjective aspect of analysis that forms the core part of many earlier efforts on evaluating the reasoning capabilities of LLMs, we automate the evaluation by leveraging models and tools from the automated planning community. The evaluation itself is done

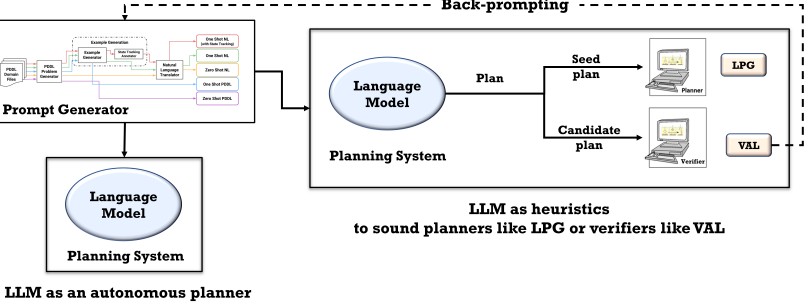

Figure 1: The diagrammatic overview of the two modes of LLMs for planning.

in two modes (shown in Figure 1). In the first "autonomous" mode, LLMs are used standalone, and we directly assess the quality and correctness of plans they generate. As we shall see, the results in the autonomous mode are pretty bleak. On an average, only about 12% of the plans that the best LLM (GPT-4) generates are actually executable without errors and reach their goals. We will show that the choice of the specific LLM (we have tested the family of GPT LLMs including GPT-4 [25], GPT-3.5 [24], InstructGPT-3.5, InstructGPT-3 [26] and GPT-3 [3]), as well as fine tuning does not seem to have a major effect on this dismal performance. We also show that the performance deteriorates further if the names of the actions and objects in the domain are obfuscated–a change that doesn't in anyway affect the performance of the standard AI planners. To shed further light on the performance of GPT4, we present an evaluation of the plans it generates under a series of more relaxed (more forgiving) executability conditions. Further, we provide a human baseline for the simplest domain in our set of domains, by presenting the planning instances to human subjects (through IRB-approved studies) and evaluating the quality and correctness of their plans. These results are *substantially better* than those of LLMs–confirming that LLMs can't plan even in a simple common sense domain in the autonomous mode. In the second "heuristic" mode, the plans produced by LLMs are given as input to an automated planner working off of a correct domain model to check whether the LLM's plans help with the search process of the underlying planner to come up with correct plans. Specifically we show that a well known automated planner called LPG [6], that uses

[2]Link to the github repo: `https://github.com/karthikv792/LLMs-Planning`

local search to locate and remove flaws in a candidate plan to make it correct, is able to repair the LLM plans with relative ease. We compare the LLM+LPG combination with two baselines, one where an empty plan is used as the seed plan for the LPG and two, where a random plan is provided as the seed plan to the LPG. We show that the average search steps by the LLM+LPG combination is much lesser than both the baselines, thereby revealing that LLMs' plans are indeed helping with the search process of the underlying planner. Further, instead of having LPG correct the plans, we use an external verifier, VAL [11], to point out the errors in the LLM-generated plans and back-prompt the LLM for a new plan with this feedback. We show that this repeated interaction indeed improves the plan correctness in common-sense domains. Overall, our findings demonstrate that, with respect to planning, LLMs' perform poorly in the autonomous mode but the generated plans can help AI planners in the search process or can be given to external verifiers and back-prompt the LLM for better plans. In this paper, we first present an overview of the related work. Following that, we describe the necessary background and the prompt generation pipeline. Finally, we provide the results and analysis of various experiments undertaken in both autonomous and heuristic evaluation modes.

## 2 Related Work

In this work, we look at LLMs' planning capabilities when the domain is given as part of the prompt (as is the standard practice in automated planning [8]). Our evaluation focuses on zero-shot (just domain and problem specification), and few-shot (example problems with plans) modes. There have been a few works that looked at the planning capabilities of LLMs. Most of them, such as [12, 2] focus on commonsense domains/tasks (e.g. moving things in kitchens, wedding/menu planning etc.) and thus evaluate LLMs in a mode wherein the prompt doesn't include any information about the specific domain. Plans generated in that way are hard to evaluate as they are not directed at any plan executor and the humans often wind up giving the benefit of doubt for a plausible–but not actually executable–plan. This is why in SayCan [2], where executability is critical, they try to filter out/interpret the LLM plans in terms of the skills/actions that are actually available to the executor. While SayCan does this in a rather convoluted way that requires access to the internal log probabilities of the LLM, our approach simplifies this by specifying the domain as part of the prompt. In all our experiments, we found that LLMs only use the actions listed as part of the domain specification.

One other mode of evaluation of planning capabilities in the literature involves the user incrementally interacting with the LLM, and re-prompting it to point out flaws in its plans, with the hope that the LLM eventually reaches an executable plan [13, 39, 28]. Such evaluations are notorious for their Clever Hans effect [1] with the actual planning being done by the humans in the loop rather than the LLMs themselves. We thus separate our evaluation into two modes–autonomous and as assistants to external planners/reasoners. There have also been efforts which mostly depended on LLMs as "translators" of natural language problem/goal specification into formal specifications, which are then thrown over to sound external planners [38, 21]. Such efforts don't shed any light on the internal planning capabilities of the LLMs themselves, as our evaluations in autonomous and assistive modes do. Finally, after our initial study and benchmark were made public, other groups did parallel studies that largely corroborate our results on the ineffectiveness of LLMs in finding executable plans [32, 21].

Taking a broader perspective, making plans in the world involves (1) discovering actions (and their precondition/effect causal dependencies), and (2) sequencing an appropriate subset of available/discovered actions to achieve the agent's goals. The former requires *broad knowledge* about actions available in the world and their individual effects, while the latter requires deep drilling-down over a given set of actions to ensure that all goals are supported (causal chaining) without any undesirable interactions. LLMs have an edge on the former–they do indeed have web-scale broad knowledge! As we shall see however, they are very bad at the second phase of developing valid interaction-free plans (in part, because LLMs don't have the ability to do combinatorial search). Most cases in literature (as outlined in [18]) where LLMs are claimed to have "planned" turn out, upon close examination, to be instances of phase 1–your wedding plans, recipe plans etc.–where you are either using a very forgiving plan correctness criterion, or the phase 2 is vacuous. Standard AI planners–on the other hand–assume that the discovery part is done and handed down as a compact domain model, and focus mostly on the second part: selecting among known actions to establish causal chains and sequencing them to make them interaction free. In this sense, LLMs and AI planners can be complementary, as we have shown in this paper–with the former helping with phase 1–either with a candidate/approximate plan or domain model–and the latter with phase 2.

# 3 Prompt Generation for Classical Planning Problems

## 3.1 Background

Given that we are interested in investigating the basic reasoning about actions and change problem, we want to look at the most fundamental planning formalism first, namely the goal-directed deterministic planning problem. Colloquially referred to as *classical planning problem*, these problem classes consist of a problem domain, an initial state and a goal state. The problem domain consists of a set of fluents which correspond to predicates with some arity and a set of actions. The state-space for the planning problem is defined by the possible truth assignment over the predicates. Each action consists of preconditions and effects where preconditions is a set of predicates that describe *when* an action can be executed and effects are set of predicates that describe *what happens* when an action is executed. The effects can further consist of add effects, which is the set of predicates that will be set true by the action, and delete effects, which is the set of predicates that will be set false. The solution for a planning problem is a sequence of actions, or a plan, that when applied in the initial state will result in a state where the goal conditions are satisfied. A standard representation to specify such kind of planning problems is the Planning Definition and Domain Language (PDDL) [22]. Below is a snippet of an action from a popular benchmark problem called Blocksworld, in PDDL. The action corresponds to picking up a block in that domain.

```
(:action pickup
  :parameters (?ob)
  :precondition (and (clear ?ob) (on-table ?ob) (arm-empty))
  :effect (and (holding ?ob) (not (clear ?ob)) (not (on-table ?ob))
              (not (arm-empty))))
```

A more detailed description on classical planning problems is provided in Appendix A.1. We now will describe how we generate the prompts that are given to the LLMs.

## 3.2 Prompt Generation

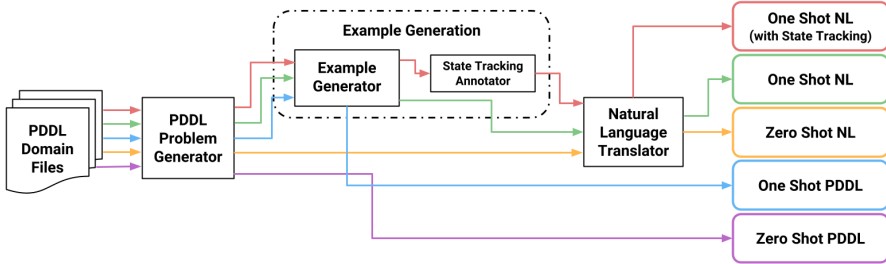

Figure 2: The diagrammatic overview of the prompt generation pipeline. The prompt configurations for the different experiments are generated from PDDL domain files and are modified with an example generator and natural language translator as needed depending on the experiment requirements.

**Prompt Configurations:** We have developed a suite of unique planning problems to test LLMs' abilities to generate plans. We have multiple prompt configurations based on this suite of problems, varying in both the method of presentation as well as number of examples given to the LLM. In particular, we use two methods of presentation, natural language and PDDL, as well as two different methods of providing examples, zero shot (with no examples provided) and one shot (with an example provided), giving us four different configuration combinations for our experiments.

Within a prompt, LLMs are first provided with a lifted domain description. For one shot configurations, the prompt additionally contains an example instance of a planning problem (consisting of a description of the initial state and the goal) and the corresponding plan (which ends with a tag, referred to as the plan-end tag, that denotes the end of the plan). All prompts end with a planning problem description. The text generated by the LLM until the plan-end tag is used as the candidate for extracting the plan. If the extractor cannot reasonably extract an instance, it is marked as incorrect.

Table 1: Results of GPT-4, GPT-3.5 (popularly known as ChatGPT), Instruct-GPT3.5, Instruct-GPT3 (text-davinci-002) and GPT3 (davinci) for the Plan Generation task with prompts in natural language.

| Domain | Method | Instances correct | | | | |
|---|---|---|---|---|---|---|
| | | GPT-4 | GPT-3.5 | I-GPT3.5 | I-GPT3 | GPT-3 |
| **Blocksworld (BW)** | One-shot | 206/600 (34.3%) | 37/600 (6.1%) | 54/600 (9%) | 41/600 (6.8%) | 6/600 (1%) |
| | Zero-shot | 210/600 (34.6%) | 8/600 (1.3%) | - | - | - |
| | COT | 214/600 (35.6%) | - | - | - | - |
| **Logistics Domain** | One-shot | 28/200 (14%) | 1/200 (0.5%) | 6/200 (3%) | 3/200 (1.5%) | - |
| | Zero-shot | 15/200 (7.5%) | 1/200 (0.5%) | - | - | - |
| **Mystery BW (Deceptive)** | One-shot | 26/600 (4.3%) | 0/600 (0%) | 4/600 (0.6%) | 14/600 (2.3%) | 0/600 (0%) |
| | Zero-shot | 1/600 (0.16%) | 0/600 (0%) | - | - | - |
| | COT | 54/600 (9%) | - | - | - | - |
| **Mystery BW (Randomized)** | One-shot | 12/600 (2%) | 0/600 (0%) | 5/600 (0.8%) | 5/600 (0.8%) | 1/600 (0.1%) |
| | Zero-shot | 0/600 (0%) | 0/600 (0%) | - | - | - |

The prompt is either formatted in natural langauge or PDDL. Natural language prompts utilize complete natural language sentences to describe feasible actions in the domain. Initial conditions are also reported as complete sentences. Plans in the natural language setting take the form of a series of commands such as "stack the orange block on top of the blue block". As implied by the name, PDDL prompts format all elements (domain description, initial state, goal state, and plans) using PDDL. We point the reader to the supplementary material for examples on each of these prompt configurations.

**Chain of Thought Prompting:** In addition to the four experiments above, we look at a fifth experiment using a state tracking chain of thought prompting technique in a natural language one shot setting. Within this configuration, we provide an annotated example where each action is annotated with the state prior to the action, the reason for why the action is applicable in the prior state, and the resulting state after applying the action. After the example, a meta-explanation about plan correctness is provided. The LLM is then asked to return a response making the same state tracking and justification annotations that were included in the example.

**Prompt Generation Pipeline:** We've developed a prompt generation pipeline (visualized in Figure 2) that accepts PDDL domain files as input and outputs prompts that follow the experiments described above. The prompt generation component takes care of creating the set of PDDL problems to be solved for all experiments. Following that, examples are added to the prompt in one shot experiments. While our setup utilizes a planner during example generation, any example generation technique could be used here so long as the examples generated are valid plans. In the state tracking experiment, we also have developed a component to add justification annotations for examples so that the examples reflect what we expect of the LLM. The last step before finishing is translation: since problems at this point are currently in PDDL, prompts for all natural language experiments (whether an example was added or not) need to be translated into natural language. We utilize a domain-specific translator to do so.

## 4 Evaluating Planning Capabilities of LLMs in Autonomous Mode

In the autonomous mode, we treat the LLM as an automated planner and perform a single run of the dataset on the LLMs for each domain and prompt configuration. In this mode, the plan generated by the LLM is back-translated from natural language to forms that can be used by external plan validators. For each domain, we perform template-based translation to translate between PDDL and natural language for the natural language prompt configurations. We use VAL [11] to evaluate the translated plan with the corresponding domain and problem file. Our evaluation here primarily

Table 2: Results of GPT-4 and GPT-3.5 (popularly known as ChatGPT) for the Plan Generation task with one or zero examples in the prompt by directly providing the domain and problem in PDDL.

| Domain | Method | Instances correct | |
| --- | --- | --- | --- |
| | | GPT-4 | GPT-3.5 |
| **Blocksworld (BW)** | One-shot | 75/600 (12.5%) | 12/600 (2%) |
| | Zero-shot | 106/600 (17.6%) | 12/600 (2%) |
| **Logistics Domain** | One-shot | 28/200 (14%) | 1/200 (0.5%) |
| | Zero-shot | 11/200 (5.5%) | 0/200 (0%) |
| **Mystery BW (Deceptive)** | One-shot | 17/600 (2.8%) | 1/600 (0.1%) |
| | Zero-shot | 3/600 (0.5%) | 0/600 (0%) |

focuses on the GPT family of LLMs. We tested GPT-4 [25] and GPT-3.5 (commonly known as Chat-GPT) [24] on all the prompt configurations while we tested the older versions of GPT (namely, Instruct-GPT3 and GPT3) on one-shot natural language prompts across the domains. We set the temperature for all models to be 0, thereby making them deterministic. In this section, we detail the evaluation of LLMs on these domains and prompt configurations. We would like to point the reader to the Appendix for example prompts.

**Evaluation of LLMs on the Blocksworld domain:** Blocksworld problems capture common sense block manipulations and consist of a set of blocks. Blocks are identified with unique colors and are placed either on a table or on top of other blocks. The goal is to arrange some of these blocks in a stack in a particular order. The general expectation here would be that one can pick up a block if it is clear, i.e., there are no other blocks on top of that block and you can only stack a block on top of another block if it is clear. The choice of this particular domain is motivated by both the fact that this is a simple common sense domain and is a very popular domain in planning literature, that has a long history of being used in various planning challenges. The instances were generated using a PDDL generator employed in the IPC competitions. We permitted the generation of problems that varied in terms of the number of blocks (3-5), optimal plan length, and goal properties (positive, negative, or no interactions between subgoals).

As shown in Table 1 and Table 2, GPT-4 improves upon previous versions of GPT models in the Blocksworld domain across all four prompt configurations. However, the overall performance is still approximately 34% in the Blocksworld dataset. Even the chain of thought style prompting (indicated by COT in the tables) had little effect on improving the performance. GPT-4 performs better with natural language prompts (206 and 210 instances for one-shot and zero-shot prompts, respectively) as opposed to PDDL prompts (75 and 106 instances). The performance drops significantly with other GPT models. We also discovered that for instances where Instruct-GPT3 generated the correct plans, replacing the example plan in the prompt with another example plan led to an even greater drop in accuracy. This suggests that the LLM seems to rely primarily on pattern matching, rather than inducing some internal model from the prompts. Overall, even in a seemingly simple common-sense domain like Blocksworld, which humans typically find easy to navigate, LLMs prove to be quite ineffective in planning autonomously.

**Finetuning GPT-3 on Blocksworld:** Along with directly testing the LLMs from the GPT family, we have also looked at the utility of fine-tuning the LLMs. Specifically, we fine-tuned GPT-3 (Davinci) in the Blocksworld domain. For this, we prepared a dataset comprising the initial state, goal state, and the respective plan for 1,000 distinct Blocksworld instances. It's important to note that these instances were separate from our test set of 600 instances. By using the default hyperparameters provided by OpenAI and an 80-20 train-validation data split, we carried out the fine-tuning process. Our results revealed that the fine-tuned GPT-3 solved only 122 instances out of the 600 in our set, representing approximately 20% of the total. This suggests that fine-tuning has a limited impact on improving the performance of LLMs in Blocksworld planning. This outcome aligns with the observations of [40], who argue that language models trained for reasoning tend to concentrate on the inherent statistical features instead of the causal structure, which in turn affects their performance on such tasks.

**Open-source models:** In addition to the GPT family of LLMs, we have also conducted preliminary experiments with an open-source LLM, BLOOM [31], and found that BLOOM too is ineffective in plan generation. We assessed BLOOM's performance in the blocksworld and mystery blocksworld (deceptive) domains using a one-shot natural language prompt configuration. In the blocksworld domain, BLOOM correctly handled only 4 out of 250 instances, representing a 1.6% success rate. In the mystery domain, it failed to produce a single correct response in all 50 instances.

**Human Baseline for the Blocksworld:** We have previously mentioned that planning tasks on the blocksworld domain are anecdotally simple enough for humans to perform. To establish this and come up with a preliminary baseline to compare LLMs performance, we conducted an IRB-approved user study where we asked 50 participants to come up with a plan for a blocksworld instance picked at random, from the set of 600 instances that we used for the evaluation of LLMs. We presented the same domain description as we did for the LLMs and then primed them with an example instance. We point the reader to the supplementary material for further details on the study.

Out of the 50 participants, 39 of them (78%) came up with a valid plan. Along with validity, we also tested the optimality of their plans even though they were not required to come up with an optimal plan. Out of the 39 participants, 35 (89.7%) participants came up with an optimal plan. These initial results show that the blocksworld domain is a simple enough domain where most humans are able to come up with plans (which are also optimal) while LLMs, on the other hand, showcase subpar performance.

**Evaluation of LLMs on the Logistics domain:** Logistics is also a widely recognized domain in the planning literature. In this domain, the objective is to transport packages within cities via trucks, and between cities via airplanes. Within a city, the locations are directly linked, allowing trucks to travel between any two of these locations. Similarly, cities are directly connected to each other allowing airplanes to travel between any two cities. Each city is equipped with one truck and has a designated location that functions as an airport. We generated 200 instances on this domain.

From Tables 1 and 2, we see that in the one-shot setting with natural language input, GPT-4 only solved 14% of the instances (28/200), and this rate dropped to 7.5% (15/200) when using zero-shot prompting. When provided with the domain and problem in PDDL format, GPT-4's performance remained the same in the one-shot setting (14% or 28/200) but decreased to 5.5% (11/200) in the zero-shot setting. GPT-3.5 did even worse.

**Obfuscating names to test the brittleness of LLM Planning:** Although the domain specification is part of our prompts, the names of the objects (e.g. blocks, trucks), predicates (e.g. on-table, in-city) and actions (e.g. pickup, drive) still do provide connections to the commonsense knowledge that the pretrained LLMs possess. One intriguing question is whether the planning performance is based really only on the domain model or these other background connections. To test this, we experimented with a variation of the Blocksworld domain, where we obfuscate the action names (for example pickup becomes attack, and unstack becomes feast) and predicate names (for example ontable becomes planet, and handempty becomes harmony). Note that from the perspective of standard planners, these domains are essentially identical.[3] In addition to such deceptive obfuscation, we also considered a variation where random alphanumeric names were substituted for the action and object names.

Tables 1 and 2, we see that this simple obfuscation leads to a catastrophic drop in performance. Specifically, with zero-shot prompting and natural language input, GPT-4 is able to solve 210 instances out of 600 in the Blocksworld domain, but it could only solve 1 instance in the deceptive Mystery Blocksworld domain and 0 instances in the randomized mystery domain. A similar result is observed with the PDDL-style prompts: GPT-4 could solve 106 instances in Blocksworld, but only 3 instances in the deceptive Mystery Blocksworld. Notably, chain of thought prompting does not significantly improve performance over one-shot natural language prompts. GPT-3.5 does not solve even a single instance in the entire set of natural language instances. For most of the instances, GPT-3.5 outputs that the instance can't be solved. These results strongly suggest that whatever accidental planning performance LLMs show is likely connected to pattern matching rather than reasoning (which should be robust to name change).

---

[3]Such obfuscated domains were originally introduced into the Intl. Planning Competition by Drew McDermott in 1998–specifically to check if the competion planners were truly domain independent.

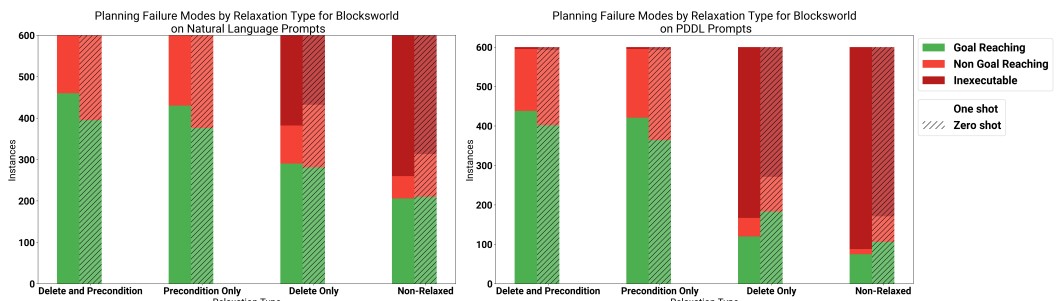

Figure 3: Assessment of GPT-4 plans with relaxations in Blocksworld domain

Table 3: Evaluation of GPT-4 and Instruct-GPT3 (I-GPT-3) plans as heuristics for a local search planner LPG, on blocksworld (BW), logistics and mystery blocksworld domains.

| Domain | LLM | Avg. Search Steps | | | Avg. Plan Length | | | Avg. Lev. Distance |
|---|---|---|---|---|---|---|---|---|
| | | Empty Seed Plan | Random Seed Plan | LLM Seed Plan | Empty Seed Plan | Random Seed Plan | LLM Seed Plan | |
| **BW** | I-GPT-3 | 15.8 | 20.07 | 14.5 | 8.45 | 9.62 | 11.7 | 7.22 |
| | GPT-4 | 15.8 | 20.07 | 8.9 | 8.45 | 9.62 | 10.76 | 4.15 |
| **Logistics** | GPT-4 | 77.5 | 144.39 | 51.3 | 23.7 | 32.72 | 32.24 | 15.04 |
| **Mystery BW** | GPT-4 | 15.8 | 20.45 | 16.09 | 8.45 | 9.78 | 11.53 | 7.77 |

**Analyzing GPT-4 failures:** To get a better sense of the type of failures LLM generated plans encounter, we wondered whether they will fare much better with a more forgiving test of the validity of the generated plans. In automated planning community, the notion of *relaxations* of the domain model are used to simplify the problem–chiefly to derive heuristics for planning problems [8]. Taking a leaf from them, we considered two types of relaxations: (i) *delete relaxation* involves ignoring all the delete conditions of the domain actions (thus making sure that there can be no negative interactions between subgoals) and (ii) *precondition relaxation* involves ignoring all the preconditions of the domain actions–thus assuming that the the actions are executable from any state giving their effects. Our idea is to evaluate the plans produced by GPT4 with respect to domain models that are delete relaxed, precondition relaxed or both. It should be clear that a plan that is correct with respect to the normal (unrelaxed) model will also be correct with respect to all the relaxed models. Figure 3 shows the results for blocksworld. We see that while the correctness of LLM generated plans increased under more forgiving (relaxed) assessments (area in green), even in the most lenient assessment mode (Delete+Precondition Relaxed), there still are plans (∼39%) that are incorrect (because they still don't reach the goals) across all the prompt configurations. The plots further classify the failure cases in terms of whether they were *inexecutable*, shown in maroon, or could be executed but didn't reach the goals (shown in red). Note that when preconditions are relaxed, all plans are executable. We provide additional details on the relaxed assessments in Appendix A.2.

## 5   Evaluating LLMs as Idea Generators

While the preceding discussion establishes that LLMs are not capable of generating correct plans in autonomous mode, there is still the possibility that they can be useful idea generators for other sound external planners, verifiers or even humans-in-the-loop. In this section, we investigate this possibility and demonstrate that LLMs show promise on this front (especially with external planners and verfiers).

### 5.1   LLM Plans as Heuristics to Sound Planners

To see if the LLM generated plans can provide heuristic guidance to sound external planners, we use a local-search planner LPG [6] which generates plans by starting with a seed plan and iteratively repairing flaws until a correct plan is found. We feed the LLM-generated plan as the initial seed plan

for LPG's iterative search. Our hypothesis is that this might put LPG on the right path and reduce the time for it to generate a correct plan. It is interesting to note the similarities between this LLM+LPG approach, and the approaches used in case-based planning in the past [9, 17]. Here the LLM can be loosely viewed as "retrieving a potentially useful plan case/sketch" out of thin air, which the LPG adapts/corrects.

We utilized the plans that were generated by LLMs in the one-shot natural language prompt configuration on all three of our previous domains - Blocksworld, Mystery Blocksworld, and Logistics - as the "seed plans" from which LPG would begin its local search for a valid plan. For the Blocksworld domain, both GPT-4 and Instruct-GPT3 were evaluated, whereas for the Logistics and Mystery domains only GPT-4 was evaluated. We confirmed that all the plans that were generated by this LLM+LPG combination for both the domains were valid (which is as expected given that the underlying planner, LPG, is sound). To get an idea of how far the initial LLM generated plans were from the final correct solutions generated by LPG, we measured the Levenshtein edit distance between them. While the default LPG local search doesn't aim to minimize the changes to the suggested plan (there do exist versions of LPG that do this; see [23]) , the edit distances also give an idea of how partially or approximately correct the original LLM plan is. Along with the edit distance, we also measured the number of search steps that were taken by the LPG to come up with a correct plan.

As shown in Table 3, the edit distances across domains are approximately half the length of the seed plans generated by the LLMs, indicating that 50% of the final plan retains the elements of the initial LLM plan. For each problem, we performed two additional plan initializations to serve as baselines: initializing with an empty plan and initializing with a random plan of the same length as the plan generated by the LLM for that problem. In the Blocksworld and Logistics[4] domains, we see a significant improvement in search steps over the empty seed plan when GPT-4 is used and an even larger one over the random seed plan. Consistent with our findings in the autonomous mode, the usefulness of this assistance wanes in domains where the relationships between predicates can no longer be inferred from common sense understandings of their names: in the Mystery Blocksworld domain, the LLM only has meager reduction in step size over the random plan and actually uses more steps than the empty plan.

## 5.2   Verifier-assisted repeated backprompting of LLMs

The interaction between LLM and LPG was unidirectional–with LLM sending a seed plan that LPG aims to repair. One supposed advantage of LLMs is that they can be prompted to improve their solutions. Suppose we have access to a sound automated verifier that not only checks the plan correctness but also pinpoints faults (in terms of unsatisfied preconditions or delete interactions). Such feedback can be easily converted into a "backprompt" to the LLM, with the hope that LLM comes up with a better plan. This is what we do with the help of VAL[11]–an AI planning tool that uses the domain model to validate the correctness of the plans (and point out errors).

Table 4: GPT4 Performance with Backprompting by VAL [11]. Mystery BW had deceptive disguising. I.C - Instances correct (within 15 feedbacks); A.F.R - Avg. feedback rounds for correct instances.

| Domain | I.C | A.F.R |
|---|---|---|
| | GPT-4 | GPT-4 |
| Blocksworld (BW) | 41/50 (82%) | 3.68 |
| Logistics | 35/50 (70%) | 3.31 |
| Mystery BW | 5/50 (10%) | 7.0 |

While, as we mentioned earlier, there can be thorny "clever hans" issues about humans prompting LLMs, an automated verifier mechanically backprompting the LLM doesn't suffer from these.

We tested this setup on a subset of the failed instances in the one-shot natural language prompt configuration using GPT-4, given its larger context window. We set a threshold of *15 backprompting rounds*. We tested on three domains–Blocksworld, Logistics and Mystery BW–with 50 failed instances from each domain. Table 4 shows the results. We provide the prompt+feedback examples in Appendix A.9. We found that GPT4 is able to come up with correct plans 82% of the Blocksworld instances and 70% of the Logistics one. The average number of backprompting rounds for these

---

[4]For the logistics domain, we considered only those instances for which a plan could be generated in less than 550 search steps (179 instances).

successful cases was 3.68 for BW and 3.31 for Logistics. The performance on the Mystery BW however remained quite poor–suggesting that even with back prompting, GPT4 cannot do well unless it can tease out commonsense patterns for the domain.

In this backprompting configuration, LLM serves as the candidate plan generator while VAL serves as the external sound verifier. While it is tempting to have a self-critiquing architecture with LLM also serving as the verifier, our recent work shows that approach to be of questionable utility as LLMs are no better at verifying plans than they are at generating them [36, 34].

### 5.3 LLMs as idea generators for humans-in-the-loop

Along with external planners and verifiers, LLMs may also offer their insights as plan suggestions directly to the human-in-the-loop which might potentially guide the user to the correct plan. After all, this sort of *computer supported cooperative work (CSCW)* use case has been the staple of LLM applications. We explored the efficacy of LLMs in assisting human planners through a *between-subjects* user study, structured similarly to the study outlined in Section 4, but with two primary distinctions: (1) The study involved two separate participant groups. The first group received no assistance in devising plans, paralleling the approach in Section 4, while the second group had access to LLM-generated suggestions. (2) both participant sets were asked to offer subjective feedback via the NASA-TLX assessment tool [10], gauging their cognitive load. Additionally, participants from the second group evaluated the correctness of the LLM suggestions presented to them. We utilized the plans generated by GPT-4 to provide plan suggestions.

The study included 49 participants in the unassisted group and 48 in the LLM-assisted group. We evaluated the statistical significance regarding accuracy, time taken, and cognitive load between the groups. Our findings revealed no statistical significance between the groups across all three aspects.[5]. Notably, 3 out of 48 participants mistakenly accepted incorrect LLM suggestions, with two submitting these erroneous suggestions as their plans. This shows the potential for automation bias in such methodologies [5]. We have provided the details of the user-study in Appendix A.12.

## 6   Conclusion and Future Work

In this paper, we presented a critical investigation of the planning abilities of large language models (LLMs). To this end, we evaluated the plan generation abilities of LLMs in two different modes. In the autonomous mode, our results show that even in simple common-sense planning domains where humans could easily come up with plans, LLMs like GPT-3 exhibit a dismal performance. Even though there is an uptick in the performance by the newer GPT-4 in the blocksworld domain, it still fails miserably on the mystery blocksworld domain, indicating their inability to reason in an abstract manner. In the heuristic mode, we have seen that plans generated by LLMs can help improve the search of sound planners like LPG. Further, we showed that using external verifiers, we can point out the errors and back-prompt LLMs for a better plan. We showed that this indeed helps in common-sense domains. In the supplementary material, we show the prompt examples for all the configurations and the details of the user-studies (Appendix A.11). From our studies, we see that LLMs as autonomous planners fail miserably, but we also see that the generated plans improve the search when used by an underlying sound planner and that better plans can be obtained by back-prompting the LLM with feedback from an external verifier.

## 7   Acknowledgements

This research was supported by ONR grants N00014-18-1-2442, N00014-18-1-2840, N00014-19-1-2119 and N00014-23-1-2409, AFOSR grant FA9550-18-1-0067, DARPA SAIL-ON grant W911NF-19-2-0006, and a JP Morgan AI Faculty Research Grant to Kambhampati. Sreedharan was supported in part by NSF grant 2303019.

---

[5]We conducted independent-samples t-tests on plan accuracy, formulation time, and task-related cognitive load between the two groups, with a significance threshold set at $\alpha$=0.05. The t-tests yielded statistic values of -1.21, 0.09, and -0.63, with p-values of 0.22, 0.92, and 0.52 for accuracy, time, and cognitive load, respectively. These results imply that the null hypothesis cannot be rejected for any of the tests, indicating no statistically significant differences in accuracy, time taken, or cognitive load between the two groups.

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
