# A Appendix

# Contents

## A.1   Classical Planning Problem Formulation

Classical Planning Problems can be mathematically represented by using the tuple $\mathcal{P} = \langle \mathcal{D}, \mathcal{I}, \mathcal{G} \rangle$. $\mathcal{D}$ is referred to as the problem domain, $I$ is the initial state and $G$ is the goal specification. The possible truth assignment over the predicates defines the state space for the planning problem. The domain is again defined by the tuple $\mathcal{D} = \langle \mathcal{F}, \mathcal{O} \rangle$. $\mathcal{F}$ corresponds to the set of fluents, i.e., the state variable used to define the state space and each fluent corresponds to a predicate with some arity, and $\mathcal{A}$ correspond to the set of actions that can be performed as part of the planning problem. Each action $a_i[\mathcal{V}] \in \mathcal{A}$ (where $a_i$ is the operator label and $\mathcal{V}$ is the variable used by the operator and each variable could be mapped to an object), can be further defined by two components, the precondition $prec[\mathcal{V}]$ which describes *when* an action can be executed and the effects $eff[\mathcal{V}]$ which defines *what happens* when an action is executed. We will assume that $prec[\mathcal{V}]$ consists of a set of predicates defined over the variables $\mathcal{V}$. An action is assumed to be executable only if its preconditions are met, i.e., the predicates in the precondition hold in the given state. The effects $eff[\mathcal{V}]$ is further defined by the tuple $\langle add[\mathcal{V}], del[\mathcal{V}] \rangle$, where $add[\mathcal{V}]$ or add effects is the set of predicates that will be set true by the action and $del[\mathcal{V}]$ or delete effects is the set of predicates that will be set false by the action. An action is said to be grounded if we replace each of the variables with an object, else it is referred to as a lifted domain model (we use a similar convention to differentiate between lifted and grounded

predicates). Below is a snippet of an action from a popular benchmark problem called Blocksworld, in PDDL. The action corresponds to picking up a block in that domain.

```
(:action pickup
  :parameters (?ob)
  :precondition (and (clear ?ob) (on-table ?ob) (arm-empty))
  :effect (and (holding ?ob) (not (clear ?ob)) (not (on-table ?ob))
              (not (arm-empty))))
```

The parameter line provides the possible variables, in this case *?ob*, which can stand for possible blocks. The precondition says that you can only pick up a block if it is clear (i.e. predicate *(clear ?ob)* is true for the block), the block is on the table and the arm is empty. The effects tell you that after you execute the action, the predicate *(holding ?ob)* becomes true and the block will no longer be considered *clear*, and *on-table*. Finally, the arm will no longer be considered *empty*. A solution to a planning problem is called a plan, and corresponds to a sequence of actions that once executed in the initial state would lead to a state where the goal specification is true. The actions may additionally be associated with cost, in these cases, one could also talk about optimal plans, i.e., a plan $\pi$ is called an optimal one if no plan exists that is less costly than $\pi$.

The above description presents one of the simpler classes of planning models and can be extended in multiple ways including allowing for object typing (including type hierarchy), more complex forms of preconditions and conditional effects, not to mention supporting richer classes of planning formalisms.

## A.2 Comparisons between the instances and plans generated by GPT-4

We have also examined the distribution of the instances (in Blocksworld and Logistics domains) that were used to test the LLMs over optimal plan lengths and the distribution of the number of correct plans by GPT-4 over the optimal plan lengths. From Figures 4, 5, 6 and 7, we can say that our traditional notions of planning complexity do not hold with LLMs. For an LLM, an easier instance from the perspective of planning complexity is the same as a harder one as it just predicts the next tokens based on their weights and the context.

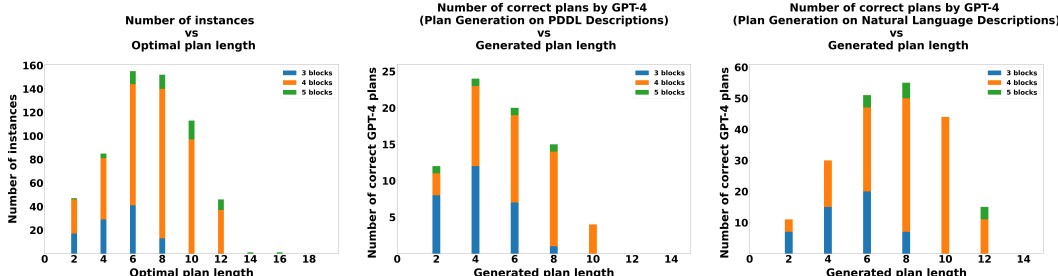

Figure 4: A detailed comparison of the blocksworld instances, against the instances where GPT-4 was able to generate a correct plan with a PDDL or a natural language style prompt which included one example.

## A.3 Failure modes

### A.3.1 LLM failures

Figure 8 shows the assessment of GPT-4 plans with relaxations in the Mystery Blocksworld domain. Similar to that of Blocksworld, there is an increase in the number of goal-reaching plans, but even in the most lenient assessment mode (Delete+Precondition Relaxation), there are quite a number of non-goal-reaching plans. In the assessment modes with precondition relaxations, an inexecutable plan is when there is an action in the plan that does not contain the required number of parameters.

Figure 9 shows the assessment of GPT-4 plans with relaxations in the Logistics domain. Even in this domain, as we further relax the assessment, we again see an increase in the number of goal reaching plans, but even the most relaxed configuration still has non-goal reaching plans.

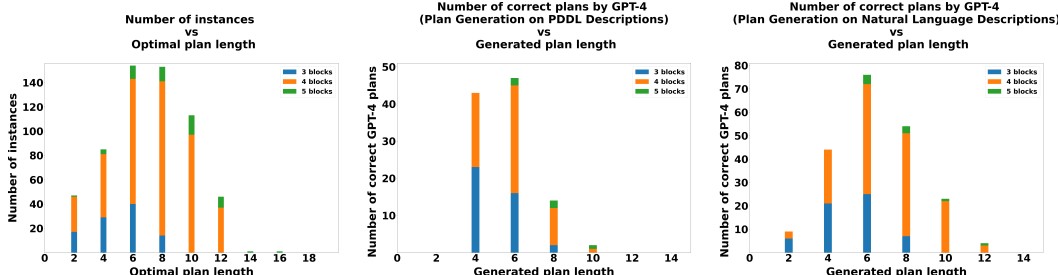

Figure 5: A detailed comparison of the blocksworld instances, against the instances where GPT-4 was able to generate a correct plan with a PDDL or a natural language style prompt which included no examples.

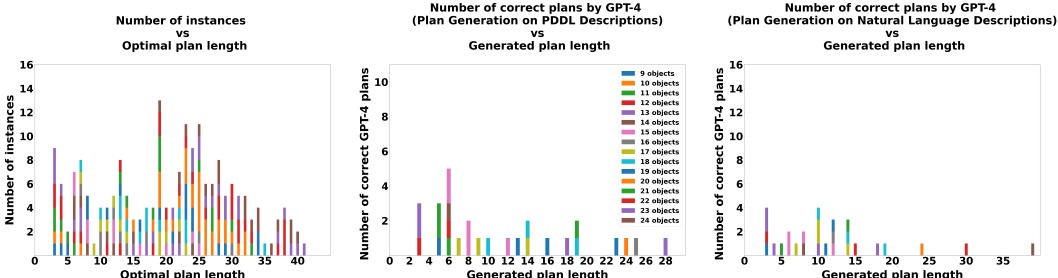

Figure 6: A detailed comparison of the logistics instances, against the instances where GPT-4 was able to generate a correct plan with a PDDL or a natural language style prompt which included one example.

### A.3.2 Human failures

For the human baseline user study (Section 4), out of 50 participants, 11 participants failed to come up with a valid plan. All the 11 participants came up with inexecutable plans. In the additional user study (in Appendix 5.3), for the first group, where the LLM assistance was not provided, out of 49 participants, 10 participants failed to come up with a valid plan and all the 10 participants came up with inexecutable plans. For the second group, where LLM assistance was provided, out of 48 participants, 15 participants failed to come up with a vaild plan out of which, 14 participants came up with an inexecutable plan and 1 participant came up with a non-goal reaching plan.

### A.4 Blocksworld Prompts in Natural Language

### A.4.1 Domain description

```
Blocksworld Domain Description

========================================
I am playing with a set of blocks where I need to arrange the blocks into stacks. Here are the
↪  actions I can do

Pick up a block
Unstack a block from on top of another block
Put down a block
Stack a block on top of another block

I have the following restrictions on my actions:
I can only pick up or unstack one block at a time.
I can only pick up or unstack a block if my hand is empty.
I can only pick up a block if the block is on the table and the block is clear. A block is clear
↪  if the block has no other blocks on top of it and if the block is not picked up.
I can only unstack a block from on top of another block if the block I am unstacking was really on
↪  top of the other block.
I can only unstack a block from on top of another block if the block I am unstacking is clear.
Once I pick up or unstack a block, I am holding the block.
```

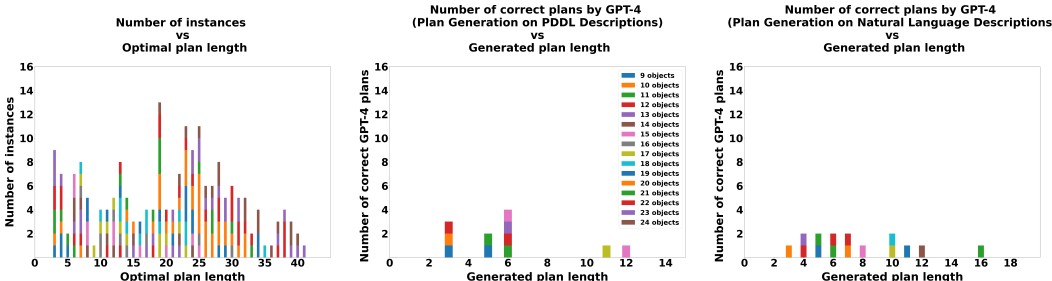

Figure 7: A detailed comparison of the logistics instances, against the instances where GPT-4 was able to generate a correct plan with a PDDL or a natural language style prompt which included no examples.

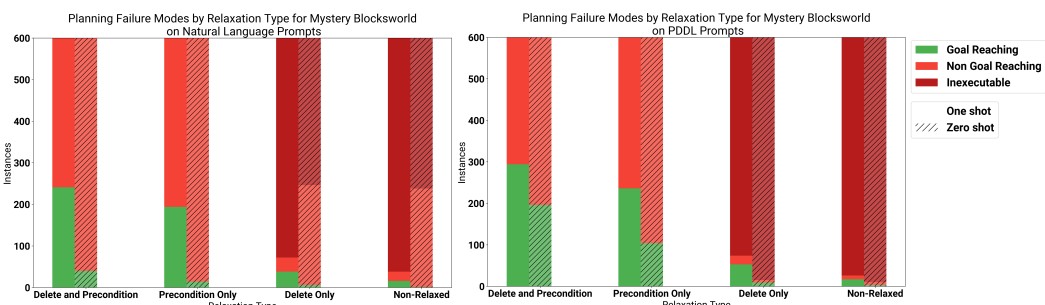

Figure 8: Assessment of GPT-4 plans with relaxations in Mystery Blocksworld (Deceptive Disguising) domain

```
I can only put down a block that I am holding.
I can only stack a block on top of another block if I am holding the block being stacked.
I can only stack a block on top of another block if the block onto which I am stacking the block
↪    is clear.
Once I put down or stack a block, my hand becomes empty.
Once you stack a block on top of a second block, the second block is no longer clear.
========================================
```

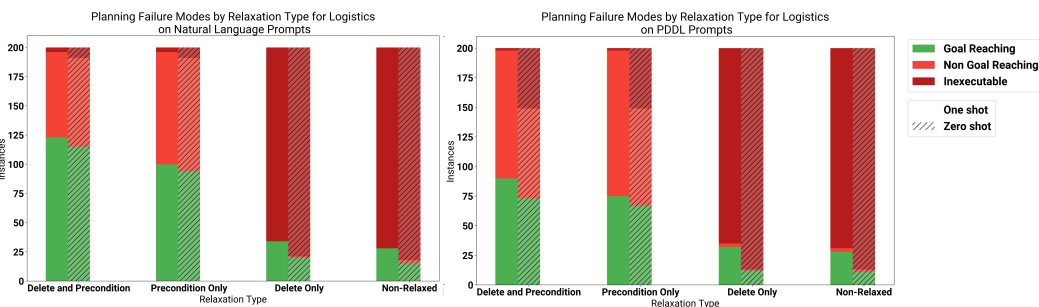

Figure 9: Assessment of GPT-4 plans with relaxations in Logistics domain

### A.4.2   One-shot prompt with GPT-4 plan

```
One-shot prompt with GPT-4 plan

[STATEMENT]
As initial conditions I have that, the red block is clear, the blue block is clear, the yellow
↪  block is clear, the hand is empty, the blue block is on top of the orange block, the red block
↪  is on the table, the orange block is on the table and the yellow block is on the table.
My goal is to have that the orange block is on top of the blue block.

My plan is as follows:

[PLAN]
unstack the blue block from on top of the orange block
put down the blue block
pick up the orange block
stack the orange block on top of the blue block
[PLAN END]

[STATEMENT]
As initial conditions I have that, the red block is clear, the yellow block is clear, the hand is
↪  empty, the red block is on top of the blue block, the yellow block is on top of the orange
↪  block, the blue block is on the table and the orange block is on the table.
My goal is to have that the orange block is on top of the red block.

My plan is as follows:

[PLAN]
--------- GPT-4 response ---------
unstack the yellow block from on top of the orange block
put down the yellow block
unstack the red block from on top of the blue block
put down the red block
pick up the orange block
stack the orange block on top of the red block
[PLAN END]
--------- Extracted plan ---------
(unstack yellow orange)
(put-down yellow)
(unstack red blue)
(put-down red)
(pick-up orange)
(stack orange red)

-------- Ground truth plan ---------
(unstack yellow orange)
(put-down yellow)
(pick-up orange)
(stack orange red)
===============SUCCESS===============
```

### A.4.3   Zero-shot prompt with GPT-4 plan

```
Zero-shot prompt with GPT-4 plan

[STATEMENT]
As initial conditions I have that, the red block is clear, the blue block is clear, the yellow
↪  block is clear, the hand is empty, the blue block is on top of the orange block, the red block
↪  is on the table, the orange block is on the table and the yellow block is on the table.
My goal is to have that the orange block is on top of the blue block.

What is the plan to achieve my goal? Just give the actions in the plan.
--------- GPT-4 response ---------
1. Unstack the blue block from on top of the orange block
2. Put down the blue block on the table
3. Pick up the orange block
4. Stack the orange block on top of the blue block
--------- Extracted plan ---------
(unstack blue orange)
(put-down blue)
(pick-up orange)
(stack orange blue)

-------- Ground truth plan ---------
(unstack blue orange)
```

```
(put-down blue)
(pick-up orange)
(stack orange blue)
===============SUCCESS===============
```

## A.4.4   State-tracking prompt with GPT-4 plan

---

**COT state-tracking prompt with GPT-4 plan**

```
The plan correctness is defined in terms of states resulting from executing the actions in the
↪   plan. An action is executable in a state when all its preconditions hold in that state. The
↪   state resulting from the action execution consists of everything in the previous state with
↪   the addition and deletion of add and delete effects of the action. Plan correctness is defined
↪   as follows: if the first action in the plan is applicable in the initial state, i.e., its
↪   preconditions are all present there; and the second action is applicable in the state
↪   resulting from applying the first action to the initial state, this process continues until
↪   the state resulting from the application of the last action in the last but one state gives
↪   rise to the final state where all the goals are satisfied.

[STATEMENT]
As initial conditions I have that, the red block is clear, the blue block is clear, the yellow
↪   block is clear, the hand is empty, the blue block is on top of the orange block, the red block
↪   is on the table, the orange block is on the table and the yellow block is on the table
My goal is to have that the orange block is on top of the blue block.
My plan is as follows:

[PLAN]
1. Current State: the red block is clear, the blue block is clear, the yellow block is clear, the
↪   hand is empty, the blue block is on top of the orange block, the red block is on the table,
↪   the orange block is on the table and the yellow block is on the table
    Action: unstack the blue block from on top of the orange block
    Reason: The above action is applicable in the current state because its preconditions; the blue
    ↪   block is clear, the hand is empty and the blue block is on top of the orange block, are
    ↪   satisfied in the current state.
    Resulting State: the red block is clear, the orange block is clear, the yellow block is clear,
    ↪   the hand is currently holding blue block, the red block is on the table, the orange block
    ↪   is on the table and the yellow block is on the table

2. Current State: the red block is clear, the orange block is clear, the yellow block is clear,
↪   the hand is currently holding blue block, the red block is on the table, the orange block is
↪   on the table and the yellow block is on the table
    Action: put down the blue block
    Reason: The above action is applicable in the current state because its preconditions; the hand
    ↪   is currently holding blue block, are satisfied in the current state.
    Resulting State: the red block is clear, the blue block is clear, the orange block is clear,
    ↪   the yellow block is clear, the hand is empty, the red block is on the table, the blue block
    ↪   is on the table, the orange block is on the table and the yellow block is on the table

3. Current State: the red block is clear, the blue block is clear, the orange block is clear, the
↪   yellow block is clear, the hand is empty, the red block is on the table, the blue block is on
↪   the table, the orange block is on the table and the yellow block is on the table
    Action: pick up the orange block
    Reason: The above action is applicable in the current state because its preconditions; the
    ↪   orange block is clear, the hand is empty and the orange block is on the table, are
    ↪   satisfied in the current state.
    Resulting State: the red block is clear, the blue block is clear, the yellow block is clear,
    ↪   the hand is currently holding orange block, the red block is on the table, the blue block
    ↪   is on the table and the yellow block is on the table

4. Current State: the red block is clear, the blue block is clear, the yellow block is clear, the
↪   hand is currently holding orange block, the red block is on the table, the blue block is on
↪   the table and the yellow block is on the table
    Action: stack the orange block on top of the blue block
    Reason: The above action is applicable in the current state because its preconditions; the blue
    ↪   block is clear and the hand is currently holding orange block, are satisfied in the current
    ↪   state.
    Resulting State: the red block is clear, the orange block is clear, the yellow block is clear,
    ↪   the hand is empty, the orange block is on top of the blue block, the red block is on the
    ↪   table, the blue block is on the table and the yellow block is on the table

Final State: the red block is clear, the orange block is clear, the yellow block is clear, the
↪   hand is empty, the orange block is on top of the blue block, the red block is on the table,
↪   the blue block is on the table and the yellow block is on the table
The goal conditions are satisfied in the final state. Hence, the above plan is valid.
[PLAN END]
```

---

[STATEMENT]
As initial conditions I have that, the red block is clear, the yellow block is clear, the hand is
↪  empty, the red block is on top of the blue block, the yellow block is on top of the orange
↪  block, the blue block is on the table and the orange block is on the table
My goal is to have that the orange block is on top of the red block.
My plan is as follows:

[PLAN]

--------- GPT-4 response ---------
1. Current State: the red block is clear, the yellow block is clear, the hand is empty, the red
↪  block is on top of the blue block, the yellow block is on top of the orange block, the blue
↪  block is on the table, and the orange block is on the table
   Action: unstack the red block from on top of the blue block
   Reason: The above action is applicable in the current state because its preconditions; the red
   ↪  block is clear, the hand is empty, and the red block is on top of the blue block, are
   ↪  satisfied in the current state.
   Resulting State: the blue block is clear, the yellow block is clear, the hand is currently
   ↪  holding red block, the yellow block is on top of the orange block, the blue block is on the
   ↪  table, and the orange block is on the table

2. Current State: the blue block is clear, the yellow block is clear, the hand is currently
↪  holding red block, the yellow block is on top of the orange block, the blue block is on the
↪  table, and the orange block is on the table
   Action: put down the red block
   Reason: The above action is applicable in the current state because the precondition; the hand
   ↪  is currently holding red block, is satisfied in the current state.
   Resulting State: the red block is clear, the blue block is clear, the yellow block is clear,
   ↪  the hand is empty, the red block is on the table, the yellow block is on top of the orange
   ↪  block, the blue block is on the table, and the orange block is on the table

3. Current State: the red block is clear, the blue block is clear, the yellow block is clear, the
↪  hand is empty, the red block is on the table, the yellow block is on top of the orange block,
↪  the blue block is on the table, and the orange block is on the table
   Action: unstack the yellow block from on top of the orange block
   Reason: The above action is applicable in the current state because its preconditions; the
   ↪  yellow block is clear, the hand is empty, and the yellow block is on top of the orange
   ↪  block, are satisfied in the current state.
   Resulting State: the red block is clear, the blue block is clear, the orange block is clear,
   ↪  the hand is currently holding yellow block, the red block is on the table, the blue block
   ↪  is on the table, and the orange block is on the table

4. Current State: the red block is clear, the blue block is clear, the orange block is clear, the
↪  hand is currently holding yellow block, the red block is on the table, the blue block is on
↪  the table, and the orange block is on the table
   Action: put down the yellow block
   Reason: The above action is applicable in the current state because the precondition; the hand
   ↪  is currently holding yellow block, is satisfied in the current state.
   Resulting State: the red block is clear, the blue block is clear, the orange block is clear,
   ↪  the yellow block is clear, the hand is empty, the red block is on the table, the blue block
   ↪  is on the table, the orange block is on the table, and the yellow block is on the table

5. Current State: the red block is clear, the blue block is clear, the orange block is clear, the
↪  yellow block is clear, the hand is empty, the red block is on the table, the blue block is on
↪  the table, the orange block is on the table, and the yellow block is on the table
   Action: pick up the orange block
   Reason: The above action is applicable in the current state because its preconditions; the
   ↪  orange block is clear, the hand is empty, and the orange block is on the table, are
   ↪  satisfied in the current state.
   Resulting State: the red block is clear, the blue block is clear, the hand is currently holding
   ↪  orange block, the yellow block is clear, the red block is on the table, the blue block is
   ↪  on the table, and the yellow block is on the table

6. Current State: the red block is clear, the blue block is clear, the hand is currently holding
↪  orange block, the yellow block is clear, the red block is on the table, the blue block is on
↪  the table, and the yellow block is on the table
   Action: stack the orange block on top of the red block
   Reason: The above action is applicable in the current state because its preconditions; the red
   ↪  block is clear, and the hand is currently holding orange block, are satisfied in the
   ↪  current state.
   Resulting State: the blue block is clear, the orange block is clear, the hand is empty, the
   ↪  yellow block is clear, the orange block is on top of the red block, the blue block is on
   ↪  the table, the red block is on the table, and the yellow block is on the table

Final State: the blue block is clear, the orange block is clear, the hand is empty, the yellow
↪  block is clear, the orange block is on top of the red block, the blue block is on the table,
↪  the red block is on the table, and the yellow block is on the table

```
The goal conditions are satisfied in the final state. Hence, the above plan is valid.
[PLAN END]
--------- Extracted plan ---------
(unstack red blue)
(put-down red)
(unstack yellow orange)
(put-down yellow)
(pick-up orange)
(stack orange red)
-------- Ground truth plan ---------
(unstack yellow orange)
(put-down yellow)
(pick-up orange)
(stack orange red)
===============SUCCESS===============
```

## A.5 Mystery Blocksworld Prompts in Natural Language

### A.5.1 Domain description (Deceptive Disguising)

### Mystery Blocksworld Domain Description (Deceptive Disguising)

```
I am playing with a set of objects. Here are the actions I can do

    Attack object
    Feast object from another object
    Succumb object
    Overcome object from another object

I have the following restrictions on my actions:
    To perform Attack action, the following facts need to be true: Province object, Planet object,
    ↪  Harmony.
    Once Attack action is performed the following facts will be true: Pain object.
    Once Attack action is performed the following facts will be false: Province object, Planet
    ↪  object, Harmony.
    To perform Succumb action, the following facts need to be true: Pain object.
    Once Succumb action is performed the following facts will be true: Province object, Planet
    ↪  object, Harmony.
    Once Succumb action is performed the following facts will be false: Pain object.
    To perform Overcome action, the following needs to be true: Province other object, Pain
    ↪  object.
    Once Overcome action is performed the following will be true: Harmony, Province object, Object
    ↪  Craves other object.
    Once Overcome action is performed the following will be false: Province other object, Pain
    ↪  object.
    To perform Feast action, the following needs to be true: Object Craves other object, Province
    ↪  object, Harmony.
    Once Feast action is performed the following will be true: Pain object, Province other object.
    Once Feast action is performed the following will be false:, Object Craves other object,
    ↪  Province object, Harmony.
```

### A.5.2 One-shot prompt with GPT-4 plan (Deceptive Disguising)

### One-shot prompt with GPT-4 plan (Deceptive Disguising)

```
[STATEMENT]
As initial conditions I have that, province object a, province object b, province object d,
↪  harmony, object b craves object c, planet object a, planet object c and planet object d.
My goal is to have that object c craves object b.

My plan is as follows:

[PLAN]
feast object b from object c
succumb object b
attack object c
overcome object c from object b
[PLAN END]

[STATEMENT]
```

```
As initial conditions I have that, province object a, province object d, harmony, object a craves
↪  object b, object d craves object c, planet object b and planet object c.
My goal is to have that object c craves object a.

My plan is as follows:

[PLAN]
--------- GPT-4 response ---------
feast object a from object b
succumb object a
attack object d
overcome object d from object a
feast object c from object a
[PLAN END]
--------- Extracted plan ---------
(feast a b)
(succumb a)
(attack d)
(overcome d a)
(feast c a)

-------- Ground truth plan ---------
(feast object d object c)
(succumb object d)
(attack object c)
(overcome object c object a)
================FAILURE================
```

### A.5.3  Zero-shot prompt with GPT-4 plan (Deceptive Disguising)

Zero-shot prompt with GPT-4 plan (Deceptive Disguising)

```
[STATEMENT]
As initial conditions I have that, object a craves object b, object d craves object c, harmony,
↪  planet object b, planet object c, province object a and province object d.
My goal is to have that object c craves object a.

What is the plan to achieve my goal? Just give the actions in the plan.
--------- GPT-4 response ---------
1. Feast object a from object b
2. Attack object a
3. Succumb object a
4. Overcome object a from object c
5. Feast object c from object a
--------- Extracted plan ---------
(feast a b)
(attack a)
(succumb a)
(overcome a c)
(feast c a)

-------- Ground truth plan ---------
(feast d c)
(succumb d)
(attack c)
(overcome c a)
================FAILURE================
```

### A.5.4  State-tracking prompt with GPT-4 plan

COT state-tracking prompt with GPT-4 plan (Deceptive Disguising)

```
The plan correctness is defined in terms of states resulting from executing the actions in the
↪  plan. An action is executable in a state when all its preconditions hold in that state. The
↪  state resulting from the action execution consists of everything in the previous state with
↪  the addition and deletion of add and delete effects of the action. Plan correctness is defined
↪  as follows: if the first action in the plan is applicable in the initial state, i.e., its
↪  preconditions are all present there; and the second action is applicable in the state
↪  resulting from applying the first action to the initial state, this process continues until
↪  the state resulting from the application of the last action in the last but one state gives
↪  rise to the final state where all the goals are satisfied.
```

[STATEMENT]
As initial conditions I have that, object b craves object c, harmony, planet object a, planet
↪ object c, planet object d, province object a, province object b and province object d
My goal is to have that object c craves object b.
My plan is as follows:

[PLAN]
1. Current State: object b craves object c, harmony, planet object a, planet object c, planet
↪ object d, province object a, province object b and province object d
   Action: feast object b from object c
   Reason: The above action is applicable in the current state because its preconditions; object b
   ↪ craves object c, harmony and province object b, are satisfied in the current state.
   Resulting State: pain object b, planet object a, planet object c, planet object d, province
   ↪ object a, province object c and province object d

2. Current State: pain object b, planet object a, planet object c, planet object d, province
↪ object a, province object c and province object d
   Action: succumb object b
   Reason: The above action is applicable in the current state because its preconditions; pain
   ↪ object b, are satisfied in the current state.
   Resulting State: harmony, planet object a, planet object b, planet object c, planet object d,
   ↪ province object a, province object b, province object c and province object d

3. Current State: harmony, planet object a, planet object b, planet object c, planet object d,
↪ province object a, province object b, province object c and province object d
   Action: attack object c
   Reason: The above action is applicable in the current state because its preconditions; harmony,
   ↪ planet object c and province object c, are satisfied in the current state.
   Resulting State: pain object c, planet object a, planet object b, planet object d, province
   ↪ object a, province object b and province object d

4. Current State: pain object c, planet object a, planet object b, planet object d, province
↪ object a, province object b and province object d
   Action: overcome object c from object b
   Reason: The above action is applicable in the current state because its preconditions; pain
   ↪ object c and province object b, are satisfied in the current state.
   Resulting State: object c craves object b, harmony, planet object a, planet object b, planet
   ↪ object d, province object a, province object c and province object d

Final State: object c craves object b, harmony, planet object a, planet object b, planet object d,
↪ province object a, province object c and province object d
The goal conditions are satisfied in the final state. Hence, the above plan is valid.
[PLAN END]

[STATEMENT]
As initial conditions I have that, object a craves object b, object d craves object c, harmony,
↪ planet object b, planet object c, province object a and province object d
My goal is to have that object c craves object a.
My plan is as follows:

[PLAN]
--------- GPT-4 response ---------
1. Current State: object a craves object b, object d craves object c, harmony, planet object b,
↪ planet object c, province object a and province object d
   Action: attack object d
   Reason: The above action is applicable in the current state because its preconditions; planet
   ↪ object d, province object d, and harmony, are satisfied in the current state.
   Resulting State: pain object d, planet object b, planet object c, province object a, province
   ↪ object b, and province object c

2. Current State: pain object d, planet object b, planet object c, province object a, province
↪ object b, and province object c
   Action: overcome object d from object a
   Reason: The above action is applicable in the current state because its preconditions; pain
   ↪ object d and province object a, are satisfied in the current state.
   Resulting State: object d craves object a, harmony, planet object b, planet object c, province
   ↪ object a, province object b, province object c, and province object d

3. Current State: object a craves object b, object d craves object a, harmony, planet object b,
↪ planet object c, province object a, province object b, province object c, and province object
↪ d
   Action: attack object b
   Reason: The above action is applicable in the current state because its preconditions; harmony,
   ↪ planet object b, and province object b, are satisfied in the current state.
   Resulting State: pain object b, planet object c, province object a, province object c, and
   ↪ province object d

```
4. Current State: pain object b, planet object c, province object a, province object c, and
↪  province object d
   Action: succumb object b
   Reason: The above action is applicable in the current state because its preconditions; pain
   ↪  object b, are satisfied in the current state.
   Resulting State: harmony, planet object c, planet object b, province object a, province object
   ↪  b, province object c, and province object d

5. Current State: harmony, planet object c, planet object b, province object a, province object b,
↪  province object c and province object d
   Action: attack object c
   Reason: The above action is applicable in the current state because its preconditions; harmony,
   ↪  planet object c, province object c, are satisfied in the current state.
   Resulting State: pain object c, planet object b, province object a, province object b, and
   ↪  province object d

6. Current State: pain object c, planet object b, province object a, province object b, and
↪  province object d
   Action: overcome object c from object a
   Reason: The above action is applicable in the current state because its preconditions; pain
   ↪  object c and province object a, are satisfied in the current state.
   Resulting State: object c craves object a, harmony, planet object b, planet object d, province
   ↪  object a, province object b and province object d

Final State: object c craves object a, harmony, planet object b, planet object d, province object
↪  a, province object b and province object d
The goal conditions are satisfied in the final state. Hence, the above plan is valid.
[PLAN END]
--------- Extracted plan ---------
(attack object d)
(overcome object d object a)
(attack object b)
(succumb object b)
(attack object c)
(overcome object c object a)

-------- Ground truth plan ---------
(feast object d object c)
(succumb object d)
(attack object c)
(overcome object c object a)
===============FAILURE===============
```

### A.5.5  Domain description (Randomized Disguising)

**Mystery Blocksworld Domain Description (Randomized Disguising)**

```
I am playing with a set of objects. Here are the actions I can do

    1jpkithdyjmlikck object
    xptxjrdkbi3pqsqr object from another object
    9big8ruzarkkquyu object
    2ijg9q8swj2shjel object from another object

I have the following restrictions on my actions:
    To perform 1jpkithdyjmlikck action, the following facts need to be true: aqcjuuehivl8auwt
    ↪  object, 51nbwlachmfartjn object, 3covmuy4yrjthijd.
    Once 1jpkithdyjmlikck action is performed the following facts will be true: gk5asm3f7u1fekpj
    ↪  object.
    Once 1jpkithdyjmlikck action is performed the following facts will be false: aqcjuuehivl8auwt
    ↪  object, 51nbwlachmfartjn object, 3covmuy4yrjthijd.
    To perform 9big8ruzarkkquyu action, the following facts need to be true: gk5asm3f7u1fekpj
    ↪  object.
    Once 9big8ruzarkkquyu action is performed the following facts will be true: aqcjuuehivl8auwt
    ↪  object, 51nbwlachmfartjn object, 3covmuy4yrjthijd.
    Once 9big8ruzarkkquyu action is performed the following facts will be false: gk5asm3f7u1fekpj
    ↪  object.
    To perform 2ijg9q8swj2shjel action, the following needs to be true: aqcjuuehivl8auwt other
    ↪  object, gk5asm3f7u1fekpj object.
    Once 2ijg9q8swj2shjel action is performed the following will be true: 3covmuy4yrjthijd,
    ↪  aqcjuuehivl8auwt object, Object 4DMF1cMTYXGSP94G other object.
    Once 2ijg9q8swj2shjel action is performed the following will be false: aqcjuuehivl8auwt other
    ↪  object, gk5asm3f7u1fekpj object.
    To perform xptxjrdkbi3pqsqr action, the following needs to be true: Object 4DMF1cMTYXGSP94G
    ↪  other object, aqcjuuehivl8auwt object, 3covmuy4yrjthijd.
```

```
      Once xptxjrdkbi3pqsqr action is performed the following will be true: gk5asm3f7u1fekpj object,
   ↪   aqcjuuehivl8auwt other object.
      Once xptxjrdkbi3pqsqr action is performed the following will be false:, Object
   ↪   4DMF1cMTYXGSP94G other object, aqcjuuehivl8auwt object, 3covmuy4yrjthijd.
```

## A.5.6   One-shot prompt with GPT-4 plan (Randomized Disguising)



One-shot prompt with GPT-4 plan (Randomized Disguising)

```
[STATEMENT]
As initial conditions I have that, aqcjuuehivl8auwt object a, aqcjuuehivl8auwt object b,
↪   aqcjuuehivl8auwt object d, 3covmuy4yrjthijd, object b 4dmf1cmtyxgsp94g object c,
↪   51nbwlachmfartjn object a, 51nbwlachmfartjn object c and 51nbwlachmfartjn object d.
My goal is to have that object c 4dmf1cmtyxgsp94g object b.

My plan is as follows:

[PLAN]
xptxjrdkbi3pqsqr object b from object c
9big8ruzarkkquyu object b
1jpkithdyjmlikck object c
2ijg9q8swj2shjel object c from object b
[PLAN END]

[STATEMENT]
As initial conditions I have that, aqcjuuehivl8auwt object a, aqcjuuehivl8auwt object d,
↪   3covmuy4yrjthijd, object a 4dmf1cmtyxgsp94g object b, object d 4dmf1cmtyxgsp94g object c,
↪   51nbwlachmfartjn object b and 51nbwlachmfartjn object c.
My goal is to have that object c 4dmf1cmtyxgsp94g object a.

My plan is as follows:

[PLAN]
--------- GPT-4 response ---------
xptxjrdkbi3pqsqr object a from object b
9big8ruzarkkquyu object a
1jpkithdyjmlikck object d
2ijg9q8swj2shjel object d from object c
[PLAN END]
--------- Extracted plan ---------
(xptxjrdkbi3pqsqr object a object b)
(9big8ruzarkkquyu object a)
(1jpkithdyjmlikck object d)
(2ijg9q8swj2shjel object d object c)

-------- Ground truth plan ---------
(xptxjrdkbi3pqsqr object d object c)
(9big8ruzarkkquyu object d)
(1jpkithdyjmlikck object c)
(2ijg9q8swj2shjel object c object a)
================FAILURE===============
```



## A.5.7   Zero-shot prompt with GPT-4 plan (Randomized Disguising)



Zero-shot prompt with GPT-4 plan (Randomized Disguising)

```
[STATEMENT]
As initial conditions I have that, aqcjuuehivl8auwt object a, aqcjuuehivl8auwt object b,
↪   aqcjuuehivl8auwt object d, 3covmuy4yrjthijd, object b 4dmf1cmtyxgsp94g object c,
↪   51nbwlachmfartjn object a, 51nbwlachmfartjn object c and 51nbwlachmfartjn object d.
My goal is to have that object c 4dmf1cmtyxgsp94g object b.

What is the plan to achieve my goal? Just give the actions in the plan.
--------- GPT3 response ---------
1. 1jpkithdyjmlikck object A
2. 2ijg9q8swj2shjel object A from object B
3. xptxjrdkbi3pqsqr object C from object A
4. 9big8ruzarkkquyu object A
--------- Extracted plan ---------
(1jpkithdyjmlikck object a)
(2ijg9q8swj2shjel object a object b)
(xptxjrdkbi3pqsqr object c object a)
```



```
(9big8ruzarkkquyu object a)

-------- Ground truth plan ---------
(xptxjrdkbi3pqsqr object b object c)
(9big8ruzarkkquyu object b)
(1jpkithdyjmlikck object c)
(2ijg9q8swj2shjel object c object b)
===============FAILURE===============
```

## A.6    Logistics Prompts in Natural Language

### A.6.1    Domain description



**Logistics Domain Description**

```
I have to plan logistics to transport packages within cities via trucks and between cities via
↪  airplanes. Locations within a city are directly connected (trucks can move between any two
↪  such locations), and so are the cities. In each city there is exactly one truck and each city
↪  has one location that serves as an airport.
Here are the actions that can be performed:

Load a package into a truck.
Load a package into an airplane.
Unload a package from a truck.
Unload a package from an airplane.
Drive a truck from one location to another location.
Fly an airplane from one city to another city.

The following are the restrictions on the actions:
A package can be loaded into a truck only if the package and the truck are in the same location.
Once a package is loaded into a truck, the package is not at the location and is in the truck.
A package can be loaded into an airplane only if the package and the airplane are in the same
↪  location.
Once a package is loaded into an airplane, the package is not at the location and is in the
↪  airplane.
A package can be unloaded from a truck only if the package is in the truck.
Once a package is unloaded from a truck, the package is not in the truck and is at the location of
↪  the truck.
A package can be unloaded from an airplane only if the package in the airplane.
Once a package is unloaded from an airplane, the package is not in the airplane and is at the
↪  location of the airplane.
A truck can be driven from one location to another if the truck is at the from-location and both
↪  from-location and to-location are locations in the same city.
Once a truck is driven from one location to another, it is not at the from-location and is at the
↪  to-location.
An airplane can be flown from one city to another if the from-location and the to-location are
↪  airports and the airplane is at the from-location.
Once an airplane is flown from one city to another the airplane is not at the from-location and is
↪  at the to-location.
```



### A.6.2    One-shot prompt with GPT-4 plan



**One-shot prompt with GPT-4 plan**

```
[STATEMENT]
As initial conditions I have that, location_0_0 is an airport, location_1_0 is an airport,
↪  airplane_0 is at location_0_0, airplane_1 is at location_0_0, package_0 is at location_1_0,
↪  truck_0 is at location_0_0, truck_1 is at location_1_0, location_0_0 is in the city city_0 and
↪  location_1_0 is in the city city_1.
My goal is to have that package_0 is at location_0_0.

My plan is as follows:

[PLAN]
fly airplane_1 from location_0_0 to location_1_0
load package_0 into airplane_1 at location_1_0
fly airplane_1 from location_1_0 to location_0_0
unload package_0 from airplane_1 at location_0_0
[PLAN END]

[STATEMENT]
```



```
As initial conditions I have that, location_0_0 is an airport, location_1_0 is an airport,
↪  airplane_0 is at location_1_0, airplane_1 is at location_1_0, package_0 is at location_1_0,
↪  truck_0 is at location_0_0, truck_1 is at location_1_0, location_0_0 is in the city city_0 and
↪  location_1_0 is in the city city_1.
My goal is to have that package_0 is at location_0_0.

My plan is as follows:

[PLAN]
--------- GPT-4 response ---------
load package_0 into airplane_1 at location_1_0
fly airplane_1 from location_1_0 to location_0_0
unload package_0 from airplane_1 at location_0_0
[PLAN END]
--------- Extracted plan ---------
(load-airplane p0 a1 l1-0)
(fly-airplane a1 l1-0 l0-0)
(unload-airplane p0 a1 l0-0)

-------- Ground truth plan ---------
(load-airplane p0 a1 l1-0)
(fly-airplane a1 l1-0 l0-0)
(unload-airplane p0 a1 l0-0)
===============SUCCESS===============
```

### A.6.3 Zero-shot prompt with GPT-4 plan

Zero-shot prompt with GPT-4 plan

```
I have to plan logistics to transport packages within cities via trucks and between cities via
↪  airplanes. Locations within a city are directly connected (trucks can move between any two
↪  such locations), and so are the cities. In each city there is exactly one truck and each city
↪  has one location that serves as an airport.
Here are the actions that can be performed:

Load a package into a truck. For example, load package_1 into truck_1 at location_1_1.
Load a package into an airplane. For example, load package_1 into airplane_1 at location_1_1.
Unload a package from a truck. For example, unload package_1 from truck_1 at location_1_1.
Unload a package from an airplane. For example, unload package_1 from airplane_1 at location_1_1.
Drive a truck from one location to another location. For example, drive truck_1 from location_1_1
↪  to location_1_2 in city_1.
Fly an airplane from one city to another city. For example, fly airplane_1 from location_1_1 to
↪  location_2_1. Here location_1_1 is the airport in city_1 and location_2_1 is the airport in
↪  city_2.

The following are the restrictions on the actions:
A package can be loaded into a truck only if the package and the truck are in the same location.
Once a package is loaded into a truck, the package is not at the location and is in the truck.
A package can be loaded into an airplane only if the package and the airplane are in the same
↪  location.
Once a package is loaded into an airplane, the package is not at the location and is in the
↪  airplane.
A package can be unloaded from a truck only if the package is in the truck.
Once a package is unloaded from a truck, the package is not in the truck and is at the location of
↪  the truck.
A package can be unloaded from an airplane only if the package in the airplane.
Once a package is unloaded from an airplane, the package is not in the airplane and is at the
↪  location of the airplane.
A truck can be driven from one location to another if the truck is at the from-location and both
↪  from-location and to-location are locations in the same city.
Once a truck is driven from one location to another, it is not at the from-location and is at the
↪  to-location.
An airplane can be flown from one city to another if the from-location and the to-location are
↪  airports and the airplane is at the from-location.
Once an airplane is flown from one city to another the airplane is not at the from-location and is
↪  at the to-location.

[STATEMENT]
As initial conditions I have that, location_0_0 is an airport, location_1_0 is an airport,
↪  airplane_0 is at location_0_0, airplane_1 is at location_0_0, package_0 is at location_1_0,
↪  truck_0 is at location_0_0, truck_1 is at location_1_0, location_0_0 is in the city city_0 and
↪  location_1_0 is in the city city_1.
My goal is to have that package_0 is at location_0_0.

What is the plan to achieve my goal? Just give the actions in the plan.
```

```
--------- GPT-4 response ---------
1. Load package_0 into truck_1 at location_1_0.
2. Drive truck_1 from location_1_0 to location_0_0 in city_1.
3. Unload package_0 from truck_1 at location_0_0.
4. Load package_0 into airplane_1 at location_0_0.
5. Fly airplane_1 from location_0_0 to location_1_0.
6. Unload package_0 from airplane_1 at location_1_0.
7. Load package_0 into truck_0 at location_1_0.
8. Fly airplane_0 from location_0_0 to location_1_0.
9. Drive truck_0 from location_1_0 to location_0_0 in city_0.
10. Unload package_0 from truck_0 at location_0_0.
--------- Extracted plan ---------
(load-truck p0 t1 l1-0)
(drive-truck t1 l1-0 l0-0 c1)
(unload-truck p0 t1 l0-0)
(load-airplane p0 a1 l0-0)
(fly-airplane a1 l0-0 l1-0)
(unload-airplane p0 a1 l1-0)
(load-truck p0 t0 l1-0)
(fly-airplane a0 l0-0 l1-0)
(drive-truck t0 l1-0 l0-0 c0)
(unload-truck p0 t0 l0-0)

-------- Ground truth plan ---------
(fly-airplane a1 l0-0 l1-0)
(load-airplane p0 a1 l1-0)
(fly-airplane a1 l1-0 l0-0)
(unload-airplane p0 a1 l0-0)
================FAILURE================
```

## A.7 Blocksworld Prompts in PDDL

### A.7.1 Domain description

---

**Blocksworld Domain Description**

```
Here is a pddl domain, an example problem and it's corresponding plan. Provide the plan for the
↪   query problem. Provide only the pddl syntax for the plan.
[DOMAIN]
(define (domain blocksworld-4ops)
  (:requirements :strips)
(:predicates (clear ?x)
             (ontable ?x)
             (handempty)
             (holding ?x)
             (on ?x ?y))

(:action pick-up
  :parameters (?ob)
  :precondition (and (clear ?ob) (ontable ?ob) (handempty))
  :effect (and (holding ?ob) (not (clear ?ob)) (not (ontable ?ob))
              (not (handempty))))

(:action put-down
  :parameters  (?ob)
  :precondition (holding ?ob)
  :effect (and (clear ?ob) (handempty) (ontable ?ob)
              (not (holding ?ob))))

(:action stack
  :parameters  (?ob ?underob)
  :precondition (and (clear ?underob) (holding ?ob))
  :effect (and (handempty) (clear ?ob) (on ?ob ?underob)
              (not (clear ?underob)) (not (holding ?ob))))

(:action unstack
  :parameters  (?ob ?underob)
  :precondition (and (on ?ob ?underob) (clear ?ob) (handempty))
  :effect (and (holding ?ob) (clear ?underob)
              (not (on ?ob ?underob)) (not (clear ?ob)) (not (handempty)))))
```

## A.7.2 One-shot prompt with GPT-4 plan

---

**One-shot prompt with GPT-4 plan**

```
[PROBLEM]
(define (problem BW-rand-4)
(:domain blocksworld-4ops)
(:objects a b c d )
(:init
(handempty)
(ontable a)
(on b c)
(ontable c)
(ontable d)
(clear a)
(clear b)
(clear d)
)
(:goal
(and
(on c b))
)
)

[PLAN]
(unstack b c)
(put-down b)
(pick-up c)
(stack c b)
[PLAN_END]

[QUERY PROBLEM]
(define (problem BW-rand-4)
(:domain blocksworld-4ops)
(:objects a b c d )
(:init
(handempty)
(on a b)
(ontable b)
(ontable c)
(on d c)
(clear a)
(clear d)
)
(:goal
(and
(on c a))
)
)

--------- GPT-4 response ---------
[PLAN]
(unstack a b)
(put-down a)
(pick-up c)
(stack c a)
[PLAN_END]
--------- Extracted plan ---------
(unstack a b)
(put-down a)
(pick-up c)
(stack c a)
-------- Ground truth plan ---------
(unstack d c)
(put-down d)
(pick-up c)
(stack c a)
===============FAILURE===============
```

### A.7.3 Zero-shot prompt with GPT-4 plan

---

**Zero-shot prompt with GPT-4 plan**

```
Here is a pddl domain, a planning problem. Provide the plan for the query problem. Provide only
↪  the pddl syntax for the plan where each action is represented as (ACTION_NAME OBJECTS).
[DOMAIN]
...
[QUERY PROBLEM]
(define (problem BW-rand-4)
(:domain blocksworld-4ops)
(:objects a b c d )
(:init
(handempty)
(ontable a)
(on b c)
(ontable c)
(ontable d)
(clear a)
(clear b)
(clear d)
)
(:goal
(and
(on c b))
)
)

--------- GPT-4 response ---------
Here is the plan for the query problem:

1. (unstack b c)
2. (put-down b)
3. (pick-up c)
4. (stack c b)
--------- Extracted plan ---------
(unstack b c)
(put-down b)
(pick-up c)
(stack c b)
-------- Ground truth plan ---------
(unstack b c)
(put-down b)
(pick-up c)
(stack c b)
===============SUCCESS===============
```

---

## A.8  Mystery Blocksworld Prompts in PDDL

### A.8.1  Domain description (Deceptive Disguising)

---

**Mystery Blocksworld Domain Description (Deceptive Disguising)**

```
Here is a pddl domain, a planning problem. Provide the plan for the query problem. Provide only
↪  the pddl syntax.
[DOMAIN]
(define (domain mystery-4ops)
  (:requirements :strips)
(:predicates (province ?x)
             (planet ?x)
             (harmony)
             (pain ?x)
             (craves ?x ?y))

(:action attack
  :parameters (?ob)
  :precondition (and (province ?ob) (planet ?ob) (harmony))
  :effect (and (pain ?ob) (not (province ?ob)) (not (planet ?ob))
               (not (harmony))))

(:action succumb
  :parameters  (?ob)
  :precondition (pain ?ob)
```

---

```
      :effect (and (province ?ob) (harmony) (planet ?ob)
                   (not (pain ?ob))))

(:action overcome
  :parameters  (?ob ?underob)
  :precondition (and (province ?underob) (pain ?ob))
  :effect (and (harmony) (province ?ob) (craves ?ob ?underob)
               (not (province ?underob)) (not (pain ?ob))))

(:action feast
  :parameters  (?ob ?underob)
  :precondition (and (craves ?ob ?underob) (province ?ob) (harmony))
  :effect (and (pain ?ob) (province ?underob)
               (not (craves ?ob ?underob)) (not (province ?ob)) (not (harmony)))))
```

### A.8.2 One-shot prompt with GPT-4 plan (Deceptive Disguising)

**One-shot prompt with GPT-4 plan (Deceptive Disguising)**

```
[PROBLEM]
(define (problem MY-rand-4)
(:domain mystery-4ops)
(:objects a b c d )
(:init
(harmony)
(planet a)
(craves b c)
(planet c)
(planet d)
(province a)
(province b)
(province d)
)
(:goal
(and
(craves c b))
)
)

[PLAN]
(feast b c)
(succumb b)
(attack c)
(overcome c b)
[PLAN_END]

[QUERY PROBLEM]
(define (problem MY-rand-4)
(:domain mystery-4ops)
(:objects a b c d )
(:init
(harmony)
(craves a b)
(planet b)
(planet c)
(craves d c)
(province a)
(province d)
)
(:goal
(and
(craves c a))
)
)

--------- GPT-4 response ---------
[QUERY_PLAN]
(feast a b)
(succumb a)
(attack c)
(overcome c a)
[QUERY_PLAN_END]
--------- Extracted plan ---------
(feast a b)
```

```
(succumb a)
(attack c)
(overcome c a)
-------- Ground truth plan ---------
(feast d c)
(succumb d)
(attack c)
(overcome c a)
===============FAILURE===============
```

### A.8.3   Zero-shot prompt with GPT-4 plan (Deceptive Disguising)

---

**Zero-shot prompt with GPT-4 plan (Deceptive Disguising)**

```
Here is a pddl domain, a planning problem. Provide the plan for the query problem. Provide only
↪  the pddl syntax for the plan where each action is represented as (ACTION_NAME OBJECTS).
[DOMAIN]
...
[QUERY PROBLEM]
(define (problem MY-rand-4)
(:domain mystery-4ops)
(:objects a b c d )
(:init
(harmony)
(planet a)
(craves b c)
(planet c)
(planet d)
(province a)
(province b)
(province d)
)
(:goal
(and
(craves c b))
)
)

--------- GPT-4 response ---------
Here is the PDDL syntax for the plan:

1. (attack a)
2. (overcome b a)
3. (feast b d)
4. (succumb d)
5. (overcome c b)
--------- Extracted plan ---------
(attack a)
(overcome b a)
(feast b d)
(succumb d)
(overcome c b)
-------- Ground truth plan ---------
(feast b c)
(succumb b)
(attack c)
(overcome c b)
===============FAILURE===============
```

---

## A.9   Logistics Prompts in PDDL

### A.9.1   Domain description

---

**Logistics Domain Description**

```
Here is a pddl domain, an example problem and it's corresponding plan. Provide the plan for the
↪  query problem. Provide only the pddl syntax for the plan.
[DOMAIN]
(define (domain logistics-strips)
  (:requirements :strips)
```

---

```
   (:predicates        (OBJ ?obj)
                       (TRUCK ?truck)
                       (LOCATION ?loc)
                 (AIRPLANE ?airplane)
                 (CITY ?city)
                 (AIRPORT ?airport)
                 (at ?obj ?loc)
                 (in ?obj1 ?obj2)
                 (in-city ?obj ?city))

  ; (:types )                ; default object

(:action LOAD-TRUCK
  :parameters
   (?obj
    ?truck
    ?loc)
  :precondition
   (and (OBJ ?obj) (TRUCK ?truck) (LOCATION ?loc)
   (at ?truck ?loc) (at ?obj ?loc))
  :effect
   (and (not (at ?obj ?loc)) (in ?obj ?truck)))

(:action LOAD-AIRPLANE
  :parameters
   (?obj
    ?airplane
    ?loc)
  :precondition
   (and (OBJ ?obj) (AIRPLANE ?airplane) (LOCATION ?loc)
   (at ?obj ?loc) (at ?airplane ?loc))
  :effect
   (and (not (at ?obj ?loc)) (in ?obj ?airplane)))

(:action UNLOAD-TRUCK
  :parameters
   (?obj
    ?truck
    ?loc)
  :precondition
   (and (OBJ ?obj) (TRUCK ?truck) (LOCATION ?loc)
        (at ?truck ?loc) (in ?obj ?truck))
  :effect
   (and (not (in ?obj ?truck)) (at ?obj ?loc)))

(:action UNLOAD-AIRPLANE
  :parameters
   (?obj
    ?airplane
    ?loc)
  :precondition
   (and (OBJ ?obj) (AIRPLANE ?airplane) (LOCATION ?loc)
        (in ?obj ?airplane) (at ?airplane ?loc))
  :effect
   (and (not (in ?obj ?airplane)) (at ?obj ?loc)))

(:action DRIVE-TRUCK
  :parameters
   (?truck
    ?loc-from
    ?loc-to
    ?city)
  :precondition
   (and (TRUCK ?truck) (LOCATION ?loc-from) (LOCATION ?loc-to) (CITY ?city)
   (at ?truck ?loc-from)
   (in-city ?loc-from ?city)
   (in-city ?loc-to ?city))
  :effect
   (and (not (at ?truck ?loc-from)) (at ?truck ?loc-to)))

(:action FLY-AIRPLANE
  :parameters
   (?airplane
    ?loc-from
    ?loc-to)
  :precondition
```

```
    (and (AIRPLANE ?airplane) (AIRPORT ?loc-from) (AIRPORT ?loc-to)
        (at ?airplane ?loc-from))
  :effect
  (and (not (at ?airplane ?loc-from)) (at ?airplane ?loc-to)))
)
```

## A.9.2   One-shot prompt with GPT-4 plan

### One-shot prompt with GPT-4 plan

```
[PROBLEM]
(define (problem logistics-c2-s1-p1-a2)
(:domain logistics-strips)
(:objects a0 a1
          c0 c1
          t0 t1
          l0-0 l1-0
          p0
)
(:init
    (AIRPLANE a0)
    (AIRPLANE a1)
    (CITY c0)
    (CITY c1)
    (TRUCK t0)
    (TRUCK t1)
    (LOCATION l0-0)
    (in-city  l0-0 c0)
    (LOCATION l1-0)
    (in-city  l1-0 c1)
    (AIRPORT l0-0)
    (AIRPORT l1-0)
    (OBJ p0)
    (at t0 l0-0)
    (at t1 l1-0)
    (at p0 l1-0)
    (at a0 l0-0)
    (at a1 l0-0)
)
(:goal
    (and
        (at p0 l0-0)
    )
)
)

[PLAN]
(fly-airplane a1 l0-0 l1-0)
(load-airplane p0 a1 l1-0)
(fly-airplane a1 l1-0 l0-0)
(unload-airplane p0 a1 l0-0)
[PLAN_END]

[QUERY PROBLEM]
(define (problem logistics-c2-s1-p1-a2)
(:domain logistics-strips)
(:objects a0 a1
          c0 c1
          t0 t1
          l0-0 l1-0
          p0
)
(:init
    (AIRPLANE a0)
    (AIRPLANE a1)
    (CITY c0)
    (CITY c1)
    (TRUCK t0)
    (TRUCK t1)
    (LOCATION l0-0)
    (in-city  l0-0 c0)
    (LOCATION l1-0)
    (in-city  l1-0 c1)
    (AIRPORT l0-0)
    (AIRPORT l1-0)
```

```
        (OBJ p0)
        (at t0 l0-0)
        (at t1 l1-0)
        (at p0 l1-0)
        (at a0 l1-0)
        (at a1 l1-0)
    )
    (:goal
        (and
            (at p0 l0-0)
        )
    )
)

--------- GPT-4 response ---------
Based on the provided domain, problem, and example plan, here is the PDDL syntax for the plan
↪  corresponding to the query problem:

(fly-airplane a0 l1-0 l0-0)
(load-airplane p0 a0 l1-0)
(fly-airplane a0 l0-0 l1-0)
(unload-airplane p0 a0 l1-0)
--------- Extracted plan ---------
(fly-airplane a0 l1-0 l0-0)
(load-airplane p0 a0 l1-0)
(fly-airplane a0 l0-0 l1-0)
(unload-airplane p0 a0 l1-0)
-------- Ground truth plan ---------
(load-airplane p0 a1 l1-0)
(fly-airplane a1 l1-0 l0-0)
(unload-airplane p0 a1 l0-0)
================FAILURE================
```

### A.9.3 Zero-shot prompt with GPT-4 plan

```
Zero-shot prompt with GPT-4 plan

Here is a pddl domain, a planning problem. Provide the plan for the query problem. Provide only
↪  the pddl syntax for the plan where each action is represented as (ACTION_NAME OBJECTS).
[DOMAIN]
...
[QUERY PROBLEM]
(define (problem logistics-c2-s1-p1-a2)
(:domain logistics-strips)
(:objects a0 a1
          c0 c1
          t0 t1
          l0-0 l1-0
          p0
)
(:init
    (AIRPLANE a0)
    (AIRPLANE a1)
    (CITY c0)
    (CITY c1)
    (TRUCK t0)
    (TRUCK t1)
    (LOCATION l0-0)
    (in-city  l0-0 c0)
    (LOCATION l1-0)
    (in-city  l1-0 c1)
    (AIRPORT l0-0)
    (AIRPORT l1-0)
    (OBJ p0)
    (at t0 l0-0)
    (at t1 l1-0)
    (at p0 l1-0)
    (at a0 l0-0)
    (at a1 l0-0)
)
(:goal
    (and
        (at p0 l0-0)
    )
```

```
)
)

--------- GPT-4 response ---------
Based on the provided domain and problem, the plan to achieve the goal state could be:

1. LOAD-AIRPLANE p0 on a0 at l1-0
2. FLY-AIRPLANE a0 from l1-0 to l0-0
3. UNLOAD-AIRPLANE p0 from a0 at l0-0

Here is the PDDL syntax for the plan:

(LOAD-AIRPLANE p0 a0 l1-0)
(FLY-AIRPLANE a0 l1-0 l0-0)
(UNLOAD-AIRPLANE p0 a0 l0-0)
--------- Extracted plan ---------
(load-airplane p0 a0 l1-0)
(fly-airplane a0 l1-0 l0-0)
(unload-airplane p0 a0 l0-0)
-------- Ground truth plan ---------
(fly-airplane a1 l0-0 l1-0)
(load-airplane p0 a1 l1-0)
(fly-airplane a1 l1-0 l0-0)
(unload-airplane p0 a1 l0-0)
===============FAILURE===============
```

## A.10   Backprompting using VAL

### A.10.1   Blocksworld example with GPT-4

> **Back-prompt with GPT-4**
>
> ```
> I am playing with a set of blocks where I need to arrange the blocks into stacks. Here are the
> ↪  actions I can do
>
> Pick up a block
> Unstack a block from on top of another block
> Put down a block
> Stack a block on top of another block
>
> I have the following restrictions on my actions:
> I can only pick up or unstack one block at a time.
> I can only pick up or unstack a block if my hand is empty.
> I can only pick up a block if the block is on the table and the block is clear. A block is clear
> ↪  if the block has no other blocks on top of it and if the block is not picked up.
> I can only unstack a block from on top of another block if the block I am unstacking was really on
> ↪  top of the other block.
> I can only unstack a block from on top of another block if the block I am unstacking is clear.
> Once I pick up or unstack a block, I am holding the block.
> I can only put down a block that I am holding.
> I can only stack a block on top of another block if I am holding the block being stacked.
> I can only stack a block on top of another block if the block onto which I am stacking the block
> ↪  is clear.
> Once I put down or stack a block, my hand becomes empty.
> Once you stack a block on top of a second block, the second block is no longer clear.
>
> [STATEMENT]
> As initial conditions I have that, the blue block is clear, the hand is empty, the blue block is
> ↪  on top of the orange block, the orange block is on top of the yellow block, the yellow block
> ↪  is on top of the red block and the red block is on the table.
> My goal is to have that the red block is on top of the orange block and the yellow block is on top
> ↪  of the red block.
>
> My plan is as follows:
>
> [PLAN]
> unstack the blue block from on top of the orange block
> put down the blue block
> unstack the orange block from on top of the yellow block
> put down the orange block
> unstack the yellow block from on top of the red block
> stack the yellow block on top of the blue block
> pick up the red block
> stack the red block on top of the orange block
> ```

```
unstack the yellow block from on top of the blue block
stack the yellow block on top of the red block
[PLAN END]

[STATEMENT]
As initial conditions I have that, the yellow block is clear, the hand is empty, the red block is
↪  on top of the orange block, the orange block is on top of the blue block, the yellow block is
↪  on top of the red block and the blue block is on the table.
My goal is to have that the red block is on top of the yellow block and the yellow block is on top
↪  of the blue block.

My plan is as follows:

[PLAN]
--------------GPT-4 Response-----------------
unstack the red block from on top of the orange block
put down the red block
unstack the orange block from on top of the blue block
stack the orange block on top of the red block
unstack the yellow block from on top of the red block
stack the yellow block on top of the blue block
pick up the red block
stack the red block on top of the yellow block
[PLAN END]
--------------VAL Feedback-----------------
The above plan is invalid.The following action at step 1 has an unmet precondition:
unstack the red block from on top of the orange block
The unmet precondition is:
the red block is clear
--------------GPT-4 Response-----------------
Apologies for the mistake. Here's the corrected plan:

[PLAN]
unstack the yellow block from on top of the red block
put down the yellow block
unstack the red block from on top of the orange block
put down the red block
unstack the orange block from on top of the blue block
stack the orange block on top of the red block
pick up the red block
stack the red block on top of the yellow block
pick up the yellow block
stack the yellow block on top of the blue block
[PLAN END]
--------------VAL Feedback-----------------
The above plan is invalid.The following action at step 7 has an unmet precondition:
pick up the red block
The unmet precondition is:
the red block is clear
--------------GPT-4 Response-----------------
Apologies for the mistake again. Here's the corrected plan:

[PLAN]
unstack the yellow block from on top of the red block
put down the yellow block
unstack the red block from on top of the orange block
put down the red block
unstack the orange block from on top of the blue block
stack the orange block on top of the red block
unstack the red block from on top of the orange block
stack the red block on top of the yellow block
pick up the yellow block
stack the yellow block on top of the blue block
[PLAN END]
--------------VAL Feedback-----------------
The above plan is invalid. The following action at step 7 has unmet preconditions:
unstack the red block from on top of the orange block
The unmet preconditions are:
the red block is clear and the red block is on top of the orange block
--------------GPT-4 Response-----------------
I apologize for the confusion. Here's the corrected plan:

[PLAN]
unstack the yellow block from on top of the red block
put down the yellow block
unstack the red block from on top of the orange block
put down the red block
unstack the orange block from on top of the blue block
```

```
put down the orange block
pick up the red block
stack the red block on top of the yellow block
pick up the yellow block
stack the yellow block on top of the blue block
[PLAN END]
--------------VAL Feedback-----------------
The above plan is invalid.The following action at step 9 has an unmet precondition:
pick up the yellow block
The unmet precondition is:
the yellow block is clear
--------------GPT-4 Response-----------------
I apologize for the mistakes in my previous plans. Here's the corrected plan:

[PLAN]
unstack the yellow block from on top of the red block
put down the yellow block
unstack the red block from on top of the orange block
put down the red block
unstack the orange block from on top of the blue block
put down the orange block
pick up the blue block
stack the blue block on top of the yellow block
pick up the yellow block
stack the yellow block on top of the red block
pick up the red block
stack the red block on top of the orange block
[PLAN END]
--------------VAL Feedback-----------------
The above plan is invalid.The following action at step 9 has an unmet precondition:
pick up the yellow block
The unmet precondition is:
the yellow block is clear
--------------GPT-4 Response-----------------
I apologize for the repeated mistakes. Here's the corrected plan:

[PLAN]
unstack the yellow block from on top of the red block
put down the yellow block
unstack the red block from on top of the orange block
put down the red block
unstack the orange block from on top of the blue block
put down the orange block
pick up the red block
stack the red block on top of the yellow block
pick up the yellow block
stack the yellow block on top of the blue block
pick up the blue block
stack the blue block on top of the orange block
[PLAN END]
--------------VAL Feedback-----------------
The above plan is invalid.The following action at step 9 has an unmet precondition:
pick up the yellow block
The unmet precondition is:
the yellow block is clear
--------------GPT-4 Response-----------------
I apologize for the confusion. Here's the corrected plan:

[PLAN]
unstack the yellow block from on top of the red block
put down the yellow block
unstack the red block from on top of the orange block
put down the red block
unstack the orange block from on top of the blue block
put down the orange block
pick up the red block
stack the red block on top of the orange block
pick up the yellow block
stack the yellow block on top of the red block
[PLAN END]
--------------VAL Feedback-----------------
The above plan is invalid. These are the unmet goal conditions:
the red block is on top of the yellow block and the yellow block is on top of the blue block
--------------GPT-4 Response-----------------
I apologize for the confusion in my previous plans. Here's the corrected plan to achieve the goal:

[PLAN]
unstack the yellow block from on top of the red block
```

```
put down the yellow block
unstack the red block from on top of the orange block
put down the red block
unstack the orange block from on top of the blue block
put down the orange block
pick up the yellow block
stack the yellow block on top of the blue block
pick up the red block
stack the red block on top of the yellow block
[PLAN END]
===============SUCCESS================
```

## A.10.2 Mystery Blocksworld example with GPT-4

```
I am playing with a set of objects. Here are the actions I can do

   Attack object
   Feast object from another object
   Succumb object
   Overcome object from another object

I have the following restrictions on my actions:
   To perform Attack action, the following facts need to be true: Province object, Planet object,
   ↪   Harmony.
   Once Attack action is performed the following facts will be true: Pain object.
   Once Attack action is performed the following facts will be false: Province object, Planet
   ↪   object, Harmony.
   To perform Succumb action, the following facts need to be true: Pain object.
   Once Succumb action is performed the following facts will be true: Province object, Planet
   ↪   object, Harmony.
   Once Succumb action is performed the following facts will be false: Pain object.
   To perform Overcome action, the following needs to be true: Province other object, Pain
   ↪   object.
   Once Overcome action is performed the following will be true: Harmony, Pain object, Object
   ↪   Craves other object.
   Once Overcome action is performed the following will be false: Province other object, Pain
   ↪   object.
   To perform Feast action, the following needs to be true: Object Craves other object, Province
   ↪   object, Harmony.
   Once Feast action is performed the following will be true: Pain object, Province other object.
   Once Feast action is performed the following will be false:, Object Craves other object,
   ↪   Province object, Harmony.

[STATEMENT]
As initial conditions I have that, object b craves object c, harmony, planet object a, planet
↪   object c, planet object d, province object a, province object b and province object d.
My goal is to have that object c craves object b.

My plan is as follows:

[PLAN]
feast object b from object c
succumb object b
attack object c
overcome object c from object b
[PLAN END]

[STATEMENT]
As initial conditions I have that, object a craves object b, object d craves object c, harmony,
↪   planet object b, planet object c, province object a and province object d.
My goal is to have that object c craves object a.

My plan is as follows:

[PLAN]
--------------GPT-4 Response-----------------
feast object a from object b
succumb object a
attack object d
overcome object d from object c
feast object c from object a
[PLAN END]
--------------VAL Feedback-----------------
```

```
The above plan is invalid.The following action at step 3 has an unmet precondition:
attack object d
The unmet precondition is:
planet object d
--------------GPT-4 Response----------------
Apologies for the oversight. Here's the corrected plan:

[PLAN]
feast object a from object b
succumb object a
attack object c
overcome object c from object d
feast object d from object c
[PLAN END]
--------------VAL Feedback----------------
The above plan is invalid.The following action at step 3 has an unmet precondition:
attack object c
The unmet precondition is:
province object c
--------------GPT-4 Response----------------
I apologize for the mistake. Here's another attempt at a valid plan:

[PLAN]
feast object a from object b
succumb object a
attack object b
overcome object b from object c
feast object c from object d
succumb object c
[PLAN END]
--------------VAL Feedback----------------
The above plan is invalid.The following action at step 4 has an unmet precondition:
overcome object b from object c
The unmet precondition is:
province object c
--------------GPT-4 Response----------------
I apologize for the repeated mistakes. Here's a new plan:

[PLAN]
feast object a from object b
succumb object a
attack object b
overcome object b from object a
feast object a from object d
succumb object a
attack object d
overcome object d from object c
[PLAN END]
--------------VAL Feedback----------------
The above plan is invalid. The following action at step 5 has unmet preconditions:
feast object a from object d
The unmet preconditions are:
object a craves object d and province object a
--------------GPT-4 Response----------------
I apologize for the errors in my previous plans. Here's a revised plan:

[PLAN]
feast object a from object b
succumb object a
attack object b
overcome object b from object a
attack object c
succumb object c
feast object c from object a
[PLAN END]
--------------VAL Feedback----------------
The above plan is invalid.The following action at step 5 has an unmet precondition:
attack object c
The unmet precondition is:
province object c
--------------GPT-4 Response----------------
I apologize for the continued mistakes. Here's another attempt at a valid plan:

[PLAN]
feast object a from object b
succumb object a
attack object b
overcome object b from object a
```

```
feast object d from object c
succumb object d
attack object c
overcome object c from object a
[PLAN END]
--------------VAL Feedback-----------------
The above plan is invalid.The following action at step 8 has an unmet precondition:
overcome object c from object a
The unmet precondition is:
province object a
--------------GPT-4 Response-----------------
I apologize for the errors in my previous plans. After reevaluating the problem, I have determined
↪   that it is not possible to achieve the goal of having object c crave object a given the
↪   initial conditions and action restrictions provided.
--------------VAL Feedback-----------------
The above plan is invalid. This is the unmet goal condition:
object c craves object a
--------------GPT-4 Response-----------------
You are correct, the goal of having object c crave object a has not been met in the previous plans.
↪   Given the initial conditions and action restrictions provided, it appears that it is not
↪   possible to achieve this goal.
--------------VAL Feedback-----------------
The above plan is invalid. This is the unmet goal condition:
object c craves object a
--------------GPT-4 Response-----------------
You are correct, the goal of having object c crave object a has not been met in the previous plans.
↪   Given the initial conditions and action restrictions provided, it appears that it is not
↪   possible to achieve this goal.
--------------VAL Feedback-----------------
The above plan is invalid. This is the unmet goal condition:
object c craves object a
--------------GPT-4 Response-----------------
You are correct, the goal of having object c crave object a has not been met in the previous plans.
↪   Given the initial conditions and action restrictions provided, it appears that it is not
↪   possible to achieve this goal.
--------------VAL Feedback-----------------
The above plan is invalid. This is the unmet goal condition:
object c craves object a
--------------GPT-4 Response-----------------
You are correct, the goal of having object c crave object a has not been met in the previous plans.
↪   Given the initial conditions and action restrictions provided, it appears that it is not
↪   possible to achieve this goal.
--------------VAL Feedback-----------------
The above plan is invalid. This is the unmet goal condition:
object c craves object a
--------------GPT-4 Response-----------------
You are correct, the goal of having object c crave object a has not been met in the previous plans.
↪   Given the initial conditions and action restrictions provided, it appears that it is not
↪   possible to achieve this goal.
--------------VAL Feedback-----------------
The above plan is invalid. This is the unmet goal condition:
object c craves object a
--------------GPT-4 Response-----------------
You are correct, the goal of having object c crave object a has not been met in the previous plans.
↪   Given the initial conditions and action restrictions provided, it appears that it is not
↪   possible to achieve this goal.
--------------VAL Feedback-----------------
The above plan is invalid. This is the unmet goal condition:
object c craves object a
--------------GPT-4 Response-----------------
You are correct, the goal of having object c crave object a has not been met in the previous plans.
↪   Given the initial conditions and action restrictions provided, it appears that it is not
↪   possible to achieve this goal.
--------------VAL Feedback-----------------
The above plan is invalid. This is the unmet goal condition:
object c craves object a
--------------GPT-4 Response-----------------
You are correct, the goal of having object c crave object a has not been met in the previous plans.
↪   Given the initial conditions and action restrictions provided, it appears that it is not
↪   possible to achieve this goal.
--------------VAL Feedback-----------------
The above plan is invalid. This is the unmet goal condition:
object c craves object a
--------------GPT-4 Response-----------------
You are correct, the goal of having object c crave object a has not been met in the previous plans.
↪   Given the initial conditions and action restrictions provided, it appears that it is not
↪   possible to achieve this goal.
===============FAILURE================
```

### A.10.3 Logistics example with GPT-4

---

**Back-prompt with GPT-4**

```
I have to plan logistics to transport packages within cities via trucks and between cities via
↪  airplanes. Locations within a city are directly connected (trucks can move between any two
↪  such locations), and so are the cities. In each city there is exactly one truck and each city
↪  has one location that serves as an airport.
Here are the actions that can be performed:

Load a package into a truck. For example, load package_1 into truck_1 at location_1_1.
Load a package into an airplane. For example, load package_1 into airplane_1 at location_1_1.
Unload a package from a truck. For example, unload package_1 from truck_1 at location_1_1.
Unload a package from an airplane. For example, unload package_1 from airplane_1 at location_1_1.
Drive a truck from one location to another location. For example, drive truck_1 from location_1_1
↪  to location_1_2 in city_1.
Fly an airplane from one city to another city. For example, fly airplane_1 from location_1_1 to
↪  location_2_1. Here location_1_1 is the airport in city_1 and location_2_1 is the airport in
↪  city_2.

The following are the restrictions on the actions:
A package can be loaded into a truck only if the package and the truck are in the same location.
Once a package is loaded into a truck, the package is not at the location and is in the truck.
A package can be loaded into an airplane only if the package and the airplane are in the same
↪  location.
Once a package is loaded into an airplane, the package is not at the location and is in the
↪  airplane.
A package can be unloaded from a truck only if the package is in the truck.
Once a package is unloaded from a truck, the package is not in the truck and is at the location of
↪  the truck.
A package can be unloaded from an airplane only if the package in the airplane.
Once a package is unloaded from an airplane, the package is not in the airplane and is at the
↪  location of the airplane.
A truck can be driven from one location to another if the truck is at the from-location and both
↪  from-location and to-location are locations in the same city.
Once a truck is driven from one location to another, it is not at the from-location and is at the
↪  to-location.
An airplane can be flown from one city to another if the from-location and the to-location are
↪  airports and the airplane is at the from-location.
Once an airplane is flown from one city to another the airplane is not at the from-location and is
↪  at the to-location.

[STATEMENT]
As initial conditions I have that, location_0_0 is an airport, location_1_0 is an airport,
↪  airplane_0 is at location_1_0, package_0 is at location_1_0, package_1 is at location_0_0,
↪  truck_0 is at location_0_0, truck_1 is at location_1_0, location_0_0 is in the city city_0 and
↪  location_1_0 is in the city city_1.
My goal is to have that package_0 is at location_1_0 and package_1 is at location_1_0.

My plan is as follows:

[PLAN]
fly airplane_0 from location_1_0 to location_0_0
load package_1 into airplane_0 at location_0_0
fly airplane_0 from location_0_0 to location_1_0
unload package_1 from airplane_0 at location_1_0
[PLAN END]

[STATEMENT]
As initial conditions I have that, location_0_0 is an airport, location_1_0 is an airport,
↪  airplane_0 is at location_1_0, package_0 is at location_1_0, package_1 is at location_1_0,
↪  truck_0 is at location_0_0, truck_1 is at location_1_0, location_0_0 is in the city city_0 and
↪  location_1_0 is in the city city_1.
My goal is to have that package_0 is at location_0_0 and package_1 is at location_0_0.

My plan is as follows:

[PLAN]
--------------GPT-4 Response-----------------
fly airplane_0 from location_1_0 to location_0_0
load package_0 into airplane_0 at location_0_0
load package_1 into airplane_0 at location_0_0
fly airplane_0 from location_0_0 to location_1_0
unload package_0 from airplane_0 at location_1_0
unload package_1 from airplane_0 at location_1_0
[PLAN END]
--------------VAL Feedback-----------------
The above plan is invalid.The following action at step 2 has an unmet precondition:
```

```
load package_0 into airplane_0 at location_0_0
The unmet precondition is:
package_0 is at location_0_0
--------------GPT-4 Response----------------
Apologies for the mistake. Here's the corrected plan:

[PLAN]
fly airplane_0 from location_1_0 to location_0_0
load package_0 into airplane_0 at location_1_0
load package_1 into airplane_0 at location_1_0
fly airplane_0 from location_0_0 to location_1_0
unload package_0 from airplane_0 at location_0_0
unload package_1 from airplane_0 at location_0_0
[PLAN END]
--------------VAL Feedback----------------
The above plan is invalid.The following action at step 2 has an unmet precondition:
load package_0 into airplane_0 at location_1_0
The unmet precondition is:
airplane_0 is at location_1_0
--------------GPT-4 Response----------------
I apologize for the mistake again. Here's the corrected plan:

[PLAN]
fly airplane_0 from location_1_0 to location_0_0
load package_0 into airplane_0 at location_0_0
load package_1 into airplane_0 at location_0_0
fly airplane_0 from location_0_0 to location_1_0
unload package_0 from airplane_0 at location_1_0
unload package_1 from airplane_0 at location_1_0
[PLAN END]
--------------VAL Feedback----------------
The above plan is invalid.The following action at step 2 has an unmet precondition:
load package_0 into airplane_0 at location_0_0
The unmet precondition is:
package_0 is at location_0_0
--------------GPT-4 Response----------------
I apologize for the confusion. Here's the corrected plan:

[PLAN]
fly airplane_0 from location_1_0 to location_0_0
load package_1 into airplane_0 at location_0_0
fly airplane_0 from location_0_0 to location_1_0
unload package_1 from airplane_0 at location_1_0
[PLAN END]
--------------VAL Feedback----------------
The above plan is invalid.The following action at step 2 has an unmet precondition:
load package_1 into airplane_0 at location_0_0
The unmet precondition is:
package_1 is at location_0_0
--------------GPT-4 Response----------------
I apologize for the repeated mistakes. Here's the corrected plan:

[PLAN]
fly airplane_0 from location_1_0 to location_0_0
load package_1 into airplane_0 at location_1_0
fly airplane_0 from location_0_0 to location_1_0
unload package_1 from airplane_0 at location_0_0
[PLAN END]
--------------VAL Feedback----------------
The above plan is invalid.The following action at step 2 has an unmet precondition:
load package_1 into airplane_0 at location_1_0
The unmet precondition is:
airplane_0 is at location_1_0
--------------GPT-4 Response----------------
I apologize once again for the mistakes. Here's the corrected plan:

[PLAN]
load package_0 into airplane_0 at location_1_0
load package_1 into airplane_0 at location_1_0
fly airplane_0 from location_1_0 to location_0_0
unload package_0 from airplane_0 at location_0_0
unload package_1 from airplane_0 at location_0_0
[PLAN END]
===============SUCCESS===============
```

## A.11 Additional experiment details

### A.11.1 LLM experiment details and the compute cost

All the experiments were run using the OpenAI API with temperature 0, making the LLMs deterministic, and all other hyperparameters to be the default ones given by the API. For GPT-4, the version we used had an 8k context window and was used between the months of March and May. The pricing of the 8k context window GPT-4 model is $0.03 for 1K tokens for the prompt and $0.06 for 1K tokens for the completion. The total cost of compute for the autonomous mode experiments on GPT-4 was $231 and the total cost for the back-prompting experiments was $149.

### A.11.2 LPG experiment details

As mentioned above, we utilized LPG in the heuristic mode to find sound plans. We specifically use LPG 1.2 implementation without a best first search fallback (so that plans are only found using the local search method) and allow for only one search restart. We use the default heuristic evaluation function and maximum number of search steps (500). If the search is restarted, an additional 50 steps can be used (bringing the maximum number on the second pass to 550). When working with the empty plan baseline, we simply do not provide an input plan. When assessing search on LLM plans, we provide the LLM plan as the input plan. For random plans, we provide a random plan of the same length as the LLM plan as the input plan.

## A.12 User study details

We ran the user studies on an online platform Prolific and paid the participants a wage of $8.12/hour for the human baseline study (described in Section 4) and $10.29/hour for the LLM+human user study (described in Section 5.3).

### A.12.1 Instructions provided to the participants

**Consent for Study:** The expected time of participation is between 25-35 minutes. You have the right not to answer any question, and to stop participation at any time. On successful completion, you will be eligible to receive $5-8 for your participation in this study. We will need to record all the responses provided by the participants during the study. Your consent to participate in this study is completely voluntary. To protect your privacy, responses from participants will never be used individually while compiling or presenting results of the study. The results of this study may be used in reports, presentations, or publications only in an aggregate form. Please enter your prolific id and click continue with the study if you agree to take part in this study.

**Study details for participants receiving LLM assistance:** In this study, you will be coming up with a plan that achieves certain goal conditions given some initial conditions.

- A plan is a sequence of actions that achieve certain goals.
- A domain consists of the actions that can be done and the restrictions on the actions.
- A problem in the specified domain will consist of the initial conditions and the goal conditions for which a plan is a solution.

You will be dealing with the blocksworld domain which consists of playing with a set of blocks where you need to arrange the blocks into stacks. You will have to come up with a plan for one blocksworld problem. You will have an AI agent that will help you in coming up with plans. This AI agent is not perfect and can make mistakes. You get a base bonus of 50 cents.

- If you come up with a successful plan your bonus compensation increases by $1.
- If your plan is unsuccessful, your bonus compensation decreases by 50 cents.
- Random plan submissions will be rejected and the bonus compensation would not be provided for such submissions.

We recommend you to have a pen and paper to aid you in visualizing the domain whenever required. We will first look at how the blocksworld domain works and what actions can you do.

**Study details for participants not receiving LLM assistance:** In this study, you will be coming up with a plan that achieves certain goal conditions given some initial conditions.

- A plan is a sequence of actions that achieve certain goals.
- A domain consists of the actions that can be done and the restrictions on the actions.
- A problem in the specified domain will consist of the initial conditions and the goal conditions for which a plan is a solution.

You will be dealing with the blocksworld domain which consists of playing with a set of blocks where you need to arrange the blocks into stacks. You will have to come up with a plan for one blocksworld problem. You get a base bonus of 50 cents.

- If you come up with a successful plan your bonus compensation increases by $1.
- If your plan is unsuccessful, your bonus compensation decreases by 50 cents.
- Random plan submissions will be rejected and the bonus compensation would not be provided for such submissions.

We recommend you to have a pen and paper to aid you in visualizing the domain whenever required. We will first look at how the blocksworld domain works and what actions can you do.

**Study details for participants in the human baseline study:** In this study, you will be coming up with a plan that achieves certain goal conditions given some initial conditions.

- A plan is a sequence of actions that achieve certain goals.
- A domain consists of the actions that can be done and the restrictions on the actions.
- A problem in the specified domain will consist of the initial conditions and the goal conditions for which a plan is a solution.

You will be dealing with the blocksworld domain which consists of playing with a set of blocks where you need to arrange the blocks into stacks. You will have to come up with a plan for one blocksworld problem. You get a base bonus of 50 cents.

- If you come up with a successful plan your bonus compensation increases by 50 cents.
- If your plan is unsuccessful, your bonus compensation decreases by 50 cents.
- Random plan submissions will be rejected and the bonus compensation would not be provided for such submissions.

We recommend you to have a pen and paper to aid you in visualizing the domain whenever required. We will first look at how the blocksworld domain works and what actions can you do.

### A.12.2 Interface of the user study

We provide the interface images at the various stages of the user studies.

### A.13 Broader Impact on using LLMs for planning

Our work relies on the use of large language models trained on large amounts of web data produced by the general public. There is significant literature on the social harms–such as the perpetuation of biases–caused by the text generated by LLMs as they are trained on unwashed web data [30, 20]. Our specific focus here is looking at additional potential harms that can be caused in the context of using LLMs for planning.

An obvious first order concern with planning is safety: LLMs can easily produce factually incorrect information which might affect the execution of generated plans in terms of correctness and safety considerations. In the autonomous mode, LLM-generated plans may simply fail, or worse, they could have detrimental side effects, such as cases where the generated plan might compromise safety by ignoring a precondition in place. Further, as shown in our results, there is no guarantee that an LLM-produced plan will achieve a goal. To mitigate these effects, plans produced by LLMs should

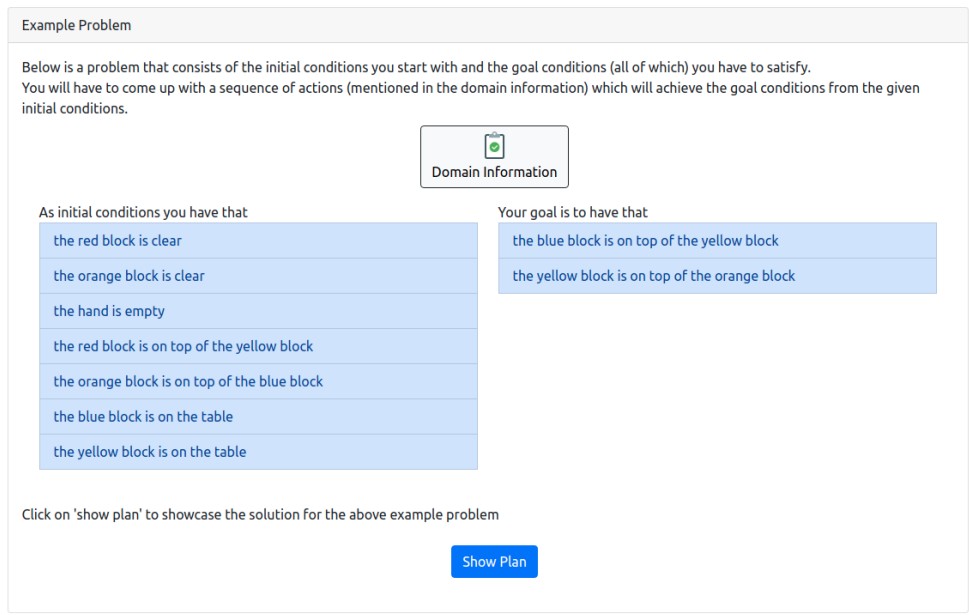

Figure 10: The description of the example problem.

be verified, which could be achieved by using either an automated verifier as in heuristic mode or a human verifier in the loop.

A subtler issue is the additional perpetuation of bias. LLMs are trained on large amounts of web data and, despite fine-tuning and training safety efforts, can take biased or implicitly harmful courses of action. For example, a wedding plan suggested by an LLM in autonomous mode might by default adhere to certain majority cultural norms. However, in our setting where we incorporate the domain model as part of the prompt, the tendency of LLMs to generate the most common or default plans is reduced if a carefully scrutinized domain model is provided.

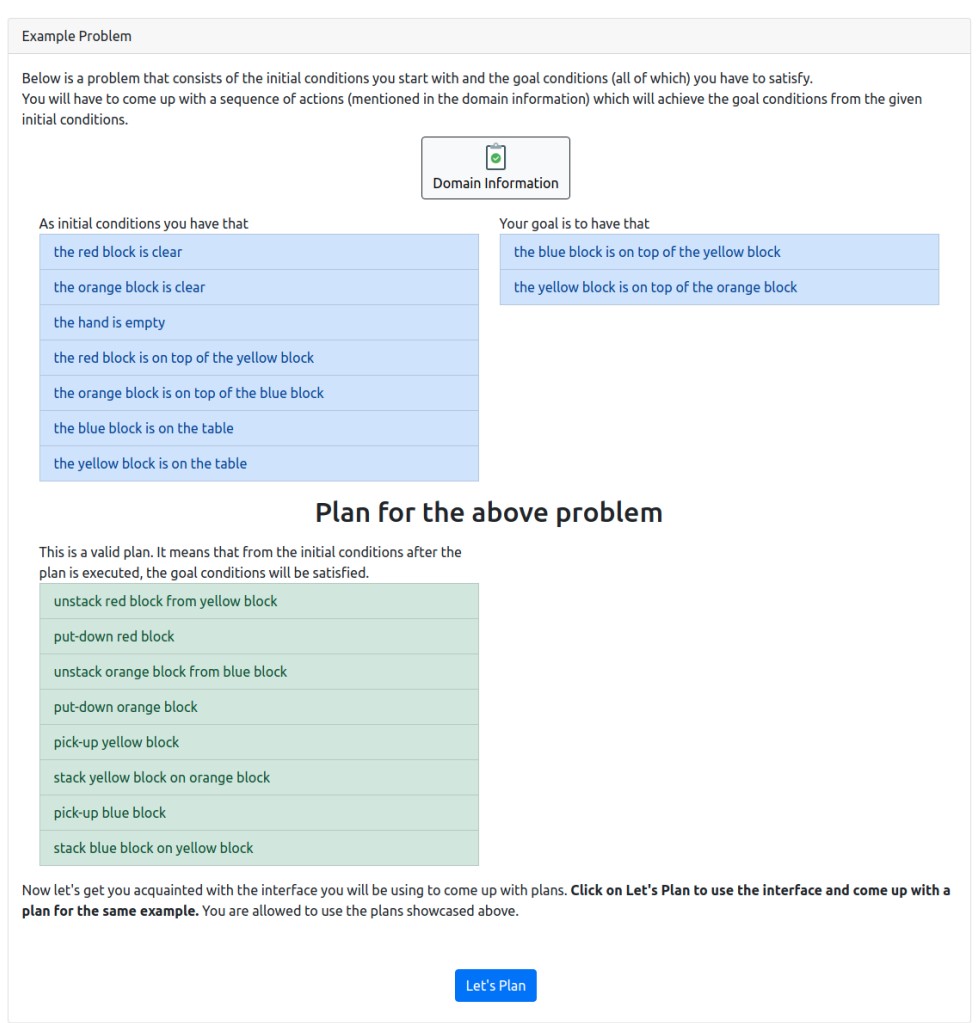

Figure 11: The description of the example problem and showcasing the solution of the example problem.

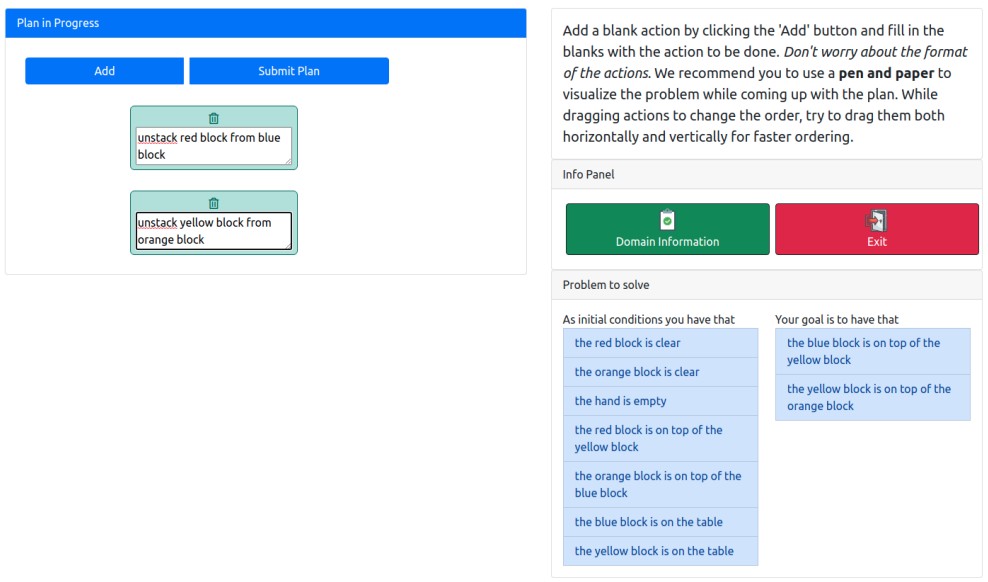

Figure 12: Interface at the plan writing phase without LLM assistance.

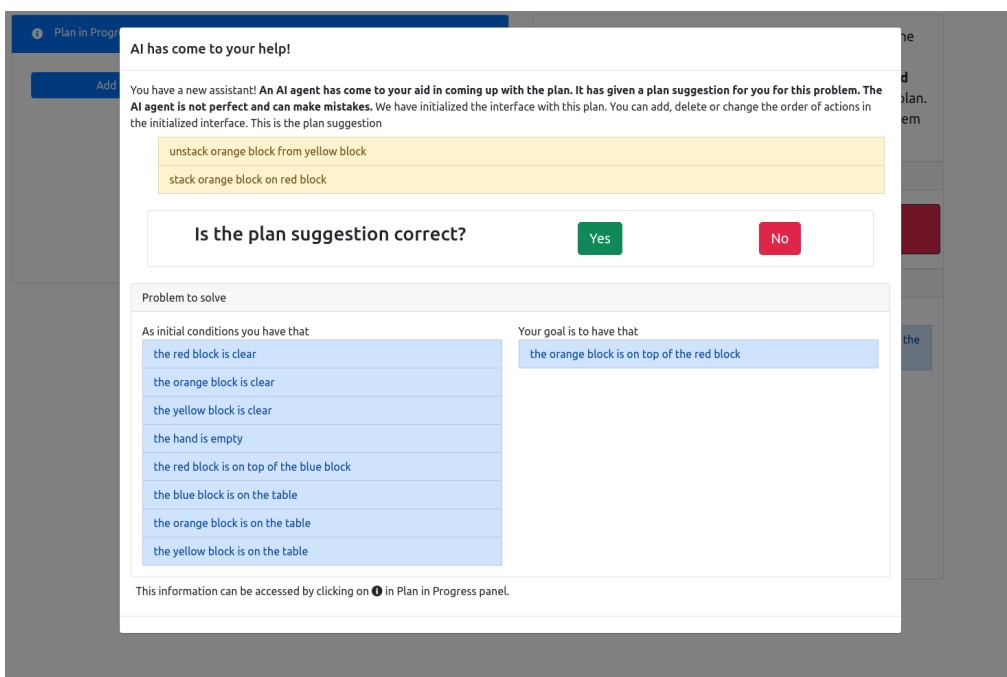

Figure 13: Interface at plan writing phase with assistance from the LLM.

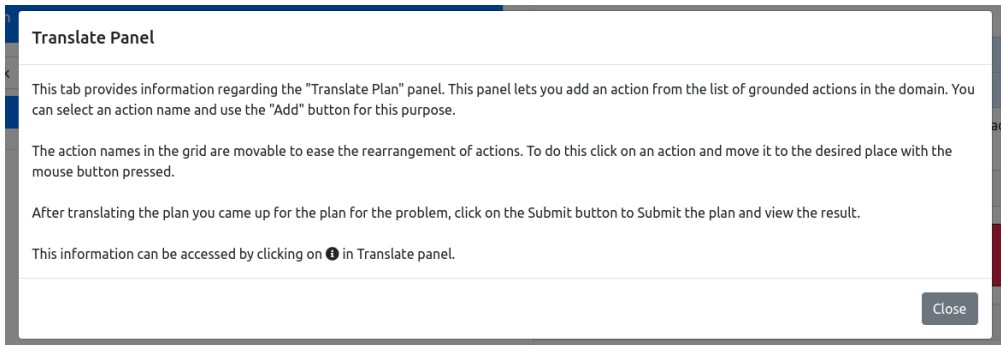

Figure 14: Description of the translate panel.

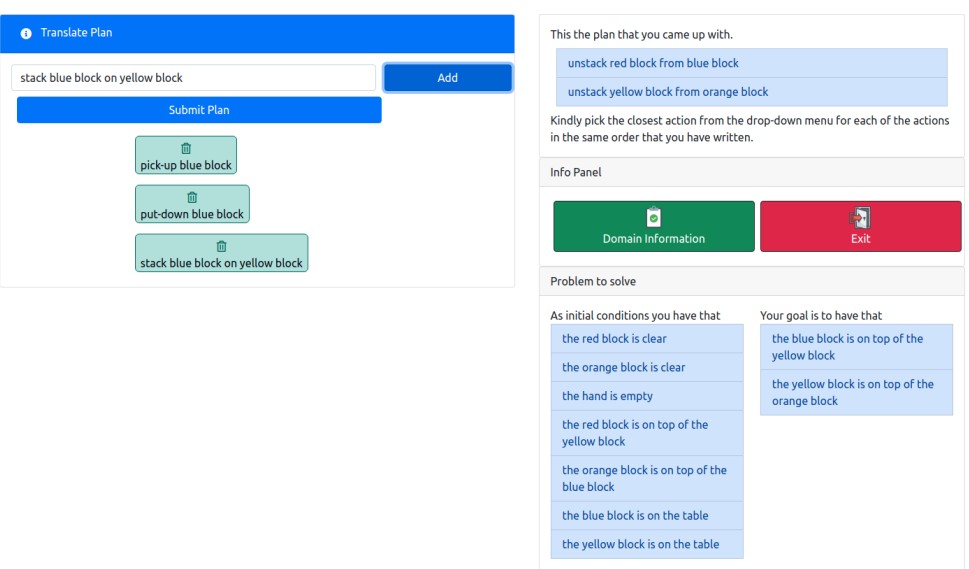

Figure 15: Interface at the plan translation phase

Please complete the below questionnaire measuring the demand of the task (only the problem instance and not the example).

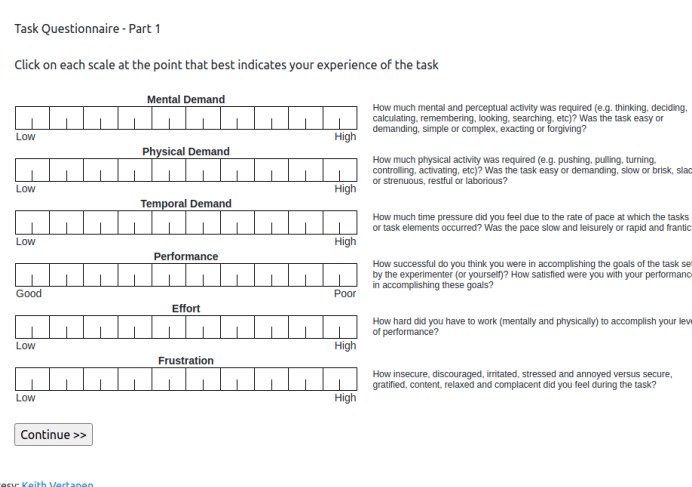

Figure 16: NASA TLX assessment at the end of the study