# OpenReview forum: "On the Planning Abilities of Large Language Models - A Critical Investigation"
_NeurIPS.cc/2023/Conference — NeurIPS 2023 spotlight_

### Official Review · Reviewer_Y3FL · 2023-07-05

**Soundness:** 2 fair
**Presentation:** 2 fair
**Contribution:** 2 fair
**Rating:** 5
**Confidence:** 3

**Summary:**

This paper conducts a systematic study by generating a suite of instances on domains similar to the ones employed in the International Planning Competition and evaluate LLMs in two distinct modes: autonomous and heuristic. The experiments show that LLMs' ability to generate executable plans autonomously is rather limited, while the results in the heuristic mode show more promise.

**Strengths:**

1. The paper is well-written and easy to understand.
2. The paper provides a detailed investigation of GPT's planning abilities in different domains and presents some interesting findings.

**Weaknesses:**

1. Some of the conclusions presented in the paper, such as re-planning makes better performance, have already been widely applied in robotics task and motion planning applications, which is not considered novel. The community has developed a range of interesting algorithms to enhance the planning capabilities of LLM, including re-plan and generating feasible plans [1,2,3,4,5]. While the authors ignores these efforts.
2. The author's investigation of GPT's planning abilities in certain domains overlooks the fact that GPT's greatest strength lies in its zero-shot or few-shot capabilities across different domains, without the need for pre-defined action spaces. Additionally, the metrics used to evaluate GPT are not fair to the model itself. At least human evaluation should be introduced for a more comprehensive assessment. More detailed experiments and metrics can be found in [zero-shot].
3. For a survey paper, it is not sufficient to only consider OpenAI's GPT-level models. Open-source models like LLaMA and Vicuna should also be included in the analysis to better understand if there are fundamental differences in planning capabilities across different levels of language models.
4. More demonstrations in the prompts seem to effectively improve planning performance. The author only conducted zero-shot and one-shot experiments, which is insufficient. A ablation experiment to explore the importance of the "how much-shot" factor would be valuable to the community.
5. Some relevant papers are not cited, which are listed in the references.
References:
[1] Inner Monologue: Embodied reasoning through planning with language models
[2] Describe, Explain, Plan and Select: Interactive Planning with Large Language Models Enables Open-World Multi-Task Agents
[3] ReAct: Synergizing Reasoning and Acting in Language Models
[4] Reflexion: Language Agents with Verbal Reinforcement Learning
[5] Text2Motion: From Natural Language Instructions to Feasible Plans

**Questions:**

Answer questions in weakness section.

---

> ### Author Rebuttal · Authors · 2023-08-10
>
> We would like to thank reviewer Y3FL for their thoughtful feedback. We are glad that the reviewer found our work to be well-written, detailed and interesting. We will incorporate all the reviewer's suggestions such as citing other relevant papers.  Below, we provide responses to some of the concerns raised by the reviewer.
>
> > 1. Some of the conclusions presented in the paper, such as re-planning makes better performance, have already been widely applied in robotics task and motion planning applications, which is not considered novel.
>
> We thank the reviewer for bringing up these relevant papers. We will make sure to cite them. Firstly, we would like to differentiate our backprompting technique from the scenarios that works like [3,4] describe. Our backprompting method provides only verification feedback that can be deduced from the original problem specification. The methods in the current works do not focus on providing verification for an end-to-end plan but instead, provide step-by-step environmental feedback that can inform later steps. This distinction is important, particularly in non-ergodic domains where irreversible actions could be executed. Additionally, the current works provide search guidance information as part of their prompts and this information is created by humans which could potentially lead to phenomena like the Clever Hans effect [1].
>
> Further, in examining the interaction resolution for the domains presented in these works, there appear to be some simplifications. In [5], the instructions seem to provide a substantial amount of the high-level plan, which could potentially reduce the LLM to a semantic parser. Similarly, in [2], the block stacking tasks often present scenarios where n-1 blocks are already stacked, only requiring the agent to stack the nth block. We believe that these efforts do not shed light on the plan generation capabilities of the LLMs themselves as both our evaluations in autonomous and heuristic modes do.
>
> [1] Clever hans or neural theory of mind? stress testing social reasoning in large language models. arXiv preprint arXiv:2305.14763.
>
> [2] Inner Monologue: Embodied reasoning through planning with language models
>
> [3] ReAct: Synergizing Reasoning and Acting in Language Models
>
> [4] Reflexion: Language Agents with Verbal Reinforcement Learning
>
> [5] Text2Motion: From Natural Language Instructions to Feasible Plans
>
> > 2. The author's investigation of GPT's planning abilities in certain domains overlooks the fact that GPT's greatest strength lies in its zero-shot or few-shot capabilities across different domains, without the need for pre-defined action spaces. Additionally, the metrics used to evaluate GPT are not fair to the model itself. At least human evaluation should be introduced for a more comprehensive assessment.
>
> As we discuss in Section 2 (lines 110-126), we readily concede that the approximate omniscience of LLMs allow them to retrieve relevant planning knowledge in many cases. The main point of our paper is that doing correct planning requires both having planning knowledge and dealing with the situation specific interaction resolution issues to ensure the correctness of the plan. Our paper shows both that LLMs can’t do the second part, and that we can gainfully leverage LLM’s approximate retrieval capabilities  in the context of external planners/verifiers to provide better planning capabilities. In other words, we are saying that LLM’s can be useful even without us erroneously ascribing them planning capabilities they don’t have.
>
> In terms of metrics, since we are talking about plan correctness, and the model is known, it makes sense to consider the categorical correctness of the plan. Human evaluations don’t provide that as humans may themselves be careless verifiers and/or suffer from automation bias (as we discuss in Appendix A.10)
>
> > 3. For a survey paper, it is not sufficient to only consider OpenAI's GPT-level models.
>
> We believe our work is not a survey paper but rather a critical examination of the planning capabilities of state-of-the-art LLMs. GPT-4 currently is the state-of-the-art among the current LLMs in natural language processing tasks. We believe that the results of GPT-4 on our planning tasks could act as an upper bound on the performance of LLMs in planning. Further, the other GPT models provide us with an approximate understanding of the planning capabilities across varying sizes of the model and fine-tuning methods (instruction-based or chat-based). We have also done preliminary experiments on BLOOM (an open-source large language model) and the results indicate bad plan generation capabilities. We have included the results in the PDF attached as part of the global response. We would also like to point out that we plan to release the required resources and code for the community to evaluate other LLMs of interest.
>
> > 4. More demonstrations in the prompts seem to effectively improve planning performance. The author only conducted zero-shot and one-shot experiments, which is insufficient. An ablation experiment to explore the importance of the "how much-shot" factor would be valuable to the community.
>
> As discussed above, our intent is not to dismiss LLM’s relevance for planning tasks, but to point out that they can be useful without us having to bend over backwards to ascribe them autonomous planning capabilities they don’t have. We won’t argue that we can’t “customize” LLM’s either by giving a large number of examples in-context or during fine-tuning–but that only increases the chance that the plan generation becomes approximate retrieval–and doesn’t prove much about the inherent plan generation capabilities of LLMs. On the whole, we believe that we have done a fair evaluation of the autonomous planning capabilities of the LLMs, giving them as much benefit of doubt as possible.

---

> > ### Comment · Reviewer_Y3FL · 2023-08-18
> >
> > Thank you for the author's reply. I hope the author can add the experimental results on other LLMs, including open-source models, to the final version.

---

### Official Review · Reviewer_GZ9T · 2023-07-07

**Soundness:** 4 excellent
**Presentation:** 3 good
**Contribution:** 4 excellent
**Rating:** 8
**Confidence:** 4

**Summary:**

This paper provides a systematic evaluation of the Planning abilities of a class of LLM (GPT series until the latest GPT-4), using standardized planning problems such as those provided in symbolic planning competitions. It analyzes LLM as autonomous planners, but also as heuristic planners providing suggestions to sound planners (heuristic mode). The experiments use a sophisticated prompting mechanism allowing both NL prompting and PDDL prompting. In the heuristic mode, LLM plans can be repaired via a local search planner (LPG), or use a verifier to send feedback to the LLM ('backprompting'). While LLMs score poorly as autonomous planners, they can achieve high scores (70-80%) in heuristic mode on standard benchmarks.

Post-rebuttal comments: the authors have fully answered my questions, in particular providing additional results for another LLM (BLOOM, which appears a very reasonable choice, due to how its ecosystem differs from that of the GPT family). Their commitment to release some of the evaluation resources can only enhance the contribution of this work, which is reflected in my increasing the 'contribution' score in this review. Having also considered other reviewers' comments and the author responses, I remain very positive about this paper and maintain my original score.

**Strengths:**

The paper introduces a comprehensive evaluation method for the Planning abilities of LLM, harnessing the full methodology of traditional Planning in terms of benchmarks and reasoners. The choice of the latter is particularly appropriate to the experiments at hand, since it includes both a local planner, well-suited to Plan repair, and a validator that can send feedback by identifying gaps or flaws in the proposed plan under the 'heuristic mode'. It is fairly impressive to have automated a process previously taking place in an interactive form with a human in the loop, which was vulnerable to the Clever Hans effect, also observed in other forms of LLM reasoning [1].
The prompting mechanism is particularly sophisticated without being over-engineered, as their is a clear rationale for supporting each option and a rather elegant design starting with PDDL domains and branching out to generate NL or PDDL prompts.
In line with some claims that LLM reasoning tend to reproduce human reasoning to some extent, the choice of planning domains known to be solvable by humans is of high interest, although some of the user/human experiments have been moved back to supplementary material.
The paper is highly readable and quite systematic, and has all the elements to become a reference paper on the topic, not least for the results produced and the contrast between autonomous and heuristic modes, the latter avoiding pitfalls of interactivity or CoT limitations.

[1] Shapira, N., Levy, M., Alavi, S.H., Zhou, X., Choi, Y., Goldberg, Y., Sap, M. and Shwartz, V., 2023. Clever hans or neural theory of mind? stress testing social reasoning in large language models. arXiv preprint arXiv:2305.14763.

**Weaknesses:**

The paper is technically sound with very few weaknesses. Perhaps one issue is the limited number of Planning test domains used in the experiments, especially compared to [Silver et al., 2022] (ref [24] in the paper). This might be in relation to the need to explore human solutions to Planning problems as described in the supplementary material, still it could be worth justifying explicitly.
Another potential issue would be concentrating on the GPT family, as LLM may vary in their real-world knowledge depending on their training base.

The paper has similarities with the following preprint: https://arxiv.org/abs/2302.06706
This is not a major issue, either in terms of novelty or in terms of anonymity, since the submission has substantial new material and there is no direct link to the authors of the preprint. Regardless of whether the preprint is from the same authors (or a subset), it would still be appropriate to reference it in the final version of the paper.


**Questions:**

Can any of the observations be related to the autoregressive nature of the LLM explored?
What variability in performance would you expect across LLMs (other than GPT versions)?

**Limitations:**

The conclusion section leaves little space to discuss limitations of the approach. Since CoT prompting has also been explored, it could have been interesting to discuss the proposed coupling of LLM to reasoners via PDDL exchange proposed as part of "Faithful CoT" [1].

The paper rightly identifies the potential role of the natural language semantics of predicates or operators' names in LLM's planning abilities, for which it designs various methods of obfuscation. However, further discussions would be interesting for this phenomenon reported in [2] ("semantics of the English terms used in the PDDL problems"), such semantics of PDDL contents having also been proposed as a mechanism for planning model extension and planning repair [3].

[1] Lyu, Q., Havaldar, S., Stein, A., Zhang, L., Rao, D., Wong, E., Apidianaki, M. and Callison-Burch, C., 2023. Faithful chain-of-thought reasoning. arXiv preprint arXiv:2301.13379.
[2] Silver, T., Hariprasad, V., Shuttleworth, R.S., Kumar, N., Lozano-Pérez, T. and Kaelbling, L.P., 2022, November. PDDL planning with pretrained large language models. In NeurIPS 2022 Foundation Models for Decision Making Workshop. - ref [24] of the paper
[3] Porteous, J., Ferreira, J.F., Lindsay, A. and Cavazza, M., 2021. Automated narrative planning model extension. Autonomous Agents and Multi-Agent Systems, 35(2), p.19.

---

> ### Author Rebuttal · Authors · 2023-08-10
>
> We would like to thank reviewer GZ9T for their detailed feedback. We are glad that the reviewer found our work to be systematic, comprehensive and likely to be a standard. We will incorporate all the reviewer's suggestions such as referencing other relevant papers and additional justifications.  Below, we provide our response to the question raised by the reviewer.
>
> > Can any of the observations be related to the autoregressive nature of the LLM explored? What variability in performance would you expect across LLMs (other than GPT versions)?
>
> Beyond the general "approximate retrieval" capabilities of the LLMs that can be attributed at some level to their auto-regressive n-grams-on-steroids nature, we did not see any other planning-specific insights. The n-gram auto-regressive nature does seem to help in the context of prompts--especially for the back-prompting techniques--in as much as it seems to get LLMs to generate the correct plan with the back-prompt augmented context. There is however no reason to believe that this is anything more than the usual context-sensitive completion capabilities.
>
> Regarding the variability of capabilities across LLMs, we believe that the results of GPT-4 could act as an upper bound on the performance of LLMs in planning as they currently are state-of-the-art in a lot of natural language processing tasks. We have also done preliminary experiments on BLOOM (an open-source large language model) and the results indicate bad plan generation capabilities. We have included the results in the PDF attached as part of the global response. We would also like to point out that we plan to release the required resources and code for the community to evaluate other LLMs of interest.

---

> > ### Comment · Reviewer_GZ9T · 2023-08-16
> >
> > Thanks for your detailed response, which answered my questions.
> > It's great to have provided additional data, in particular on BLOOM, which differs sufficiently from GPT to broaden the argument.
> > At some point, it might be interesting to investigate BLOOM's worst performance, but that is beyond the scope of this paper.

---

### Official Review · Reviewer_JKYr · 2023-07-07

**Soundness:** 4 excellent
**Presentation:** 4 excellent
**Contribution:** 3 good
**Rating:** 8
**Confidence:** 4

**Summary:**

This work evaluates the planning abilities of LLMs in two distinct settings:  (1) As generators of final plans, with or without feedback from a validator, and (2) as generators of seed plans which are then corrected by a standard planner. The evaluations are performed on two commonsense domains for which humans tend to produce high-quality plans:  Blocksworld and Logistics. Four LLMs are tested, including GPT-4, which generates more correct plans than the other models. Still, GPT-4 is found to fail on most of the problems, even with the benefit of one-shot prompting and CoT reasoning. But feedback from a validator (VAL) dramatically boosts the solution rate to 82% in BW and 70% in logistics, after just 3-4 feedback loops on average. In the other setting, the standard planner (LPG) produces correct plans in significantly fewer steps when starting with seed plans generated by GPT-4.

**Strengths:**

This is excellent work, carefully detailed, and clearly presented. It avoids the Clever Hans effect that often arises when humans are involved in evaluations.

The different evaluation settings are very well chosen, and the results provide valuable guidance as industries work to understand how these LLMs can best be leveraged.

GPT-4 is thoroughly evaluated on all benchmarks. Inclusion of three other LLMs on many of the evaluations provides additional insight.


**Weaknesses:**

Covering more domains beyond these two would be a useful contribution. But two are sufficient to support the conclusions drawn, and they light the way for others to run similar evaluations on additional planning domains.

**Questions:**

Line 97 says that “our approach of specifying the domain as part of the prompt ensures that the generated plans only use the actions in the domain specification.” How does domain specification provide this guarantee? Can't the LLM still hallucinate nonsense?

Line 191 says:  “We set the temperature for all models to be 1, thereby making them deterministic.” Is this a typo? The appendix correctly identifies zero as the temperature that produces deterministic behavior.


**Limitations:**

No concerns.

---

> ### Author Rebuttal · Authors · 2023-08-10
>
> We thank reviewer JKYr for their valuable comments. We are glad that the reviewer found our work to be detailed and well-presented. Below, we provide responses to the questions raised by the reviewer.
> > Line 97 says that “our approach of specifying the domain as part of the prompt ensures that the generated plans only use the actions in the domain specification.” How does domain specification provide this guarantee? Can't the LLM still hallucinate nonsense?
>
> We agree with the reviewer that even after specifying the actions in the domain, there is a possibility that LLMs could hallucinate the actions in the generated plan. However, in our experiments, we found that none of the LLMs hallucinates actions for any of the instances. We will update the paper to make this clearer.
> > Line 191 says: “We set the temperature for all models to be 1, thereby making them deterministic.” Is this a typo? The appendix correctly identifies zero as the temperature that produces deterministic behavior.
>
> Yes. It is a typo. We will update the paper and fix it.

---

> > ### Comment · Reviewer_JKYr · 2023-08-18
> > **Response to rebuttal**
> >
> > Thank you for the clarifications!

---

### Official Review · Reviewer_NTC1 · 2023-07-10

**Soundness:** 3 good
**Presentation:** 4 excellent
**Contribution:** 4 excellent
**Rating:** 8
**Confidence:** 5

**Summary:**

The paper investigates the (lack of) capabilities of pretrained LLM for solving classical well-known planning benchmarks. They study both the case of fully autonomous LLM without any external feedback and the case of using external tools, for validation feedback or as a seed for improving an external planner. As no fine-tuning is done, the work involves variations of prompts for the task considered. Fig 2 is an excellent summary of the studied tasks including zero and one-shot tasks with the problem in NL or in the original PDDL. GPT-4 is reported as the most powerful LLM, but it is still not satisfactory. Moreover, it's shown to be sensitive to the name description, performing better when using the standard names for well-known benchmarks.


**Strengths:**


- Some LLMs such as GPT-4 are being used to obtain plans, so it's important to investigate their capabilities.
- Classical planning benchmarks are well-understood so offer a solid ground for evaluation.
- Consider explicitly the case of autonomous mode vs external but automatic feedback.
- Chain of thought is investigated, answering a question that people familiar with LLMs might have.


**Weaknesses:**

- LLMs are trained in language and human-written code. The evaluation with Randomized Disguising might be less meaningful.
	- However, Randomized Disguising is a small part of the work.
	- An alternative not explored in the paper is to add human-readable description to the domains. Even though that requires human intervention, it's reasonable to assume the ones providing the domain can also provide a description.
- The domain-specific translator is not discussed.


**Questions:**

- Did you try COT with PDDL prompts?
- Are there any indications of how LLM would perform in problems with shallow plans? If the number of objects or actions is big enough, that might be challenging for classical planners.
- For the interactive scenario, does it make sense to keep the temperature at 0? Perhaps randomization might help the LLM to recover to deviate from earlier commitments.
- Did you attempt relaxations with other planning problems? Perhaps it's not "natural" in blocks world, but there are other problems where the plans are equivalent to their relaxed versions.


**Limitations:**

A potential limitation of this work is that it might inform about the relevant problems that required planning from LLMs. That's not the scope of the work, so that argument should be left aside. Instead, this work is a systematic investigation of well-known planning benchmarks. Those problems might be close to the distribution of the LLMs.

I miss a discussion on the complexity of the planning tasks per se. There are some simple algorithms for solving block world problems. While logistics can be a complex problem, in the PDDL benchmark solving a single problem is bounded by the complexity of moving one package, a simple problem that as cities are fully connected and it takes one airplane trip for a package to the right city. The relaxed plans discussed in the paper might be related to my question about shallow plans.

Other comments:
- Fig 2 summarizes well the approaches studied, but that's sometimes not mentioned in the table captions. For instance, Table 1 and 2 should mention that they consider the automated approach.
- What's I-GPT-3 in Table 3?

---

> ### Author Rebuttal · Authors · 2023-08-10
>
> We would like to thank reviewer NTC1 for their valuable comments. We are glad that the reviewer found our work to be important and comprehensive. Below, we provide responses to the concerns raised by the reviewer.
>
> > 1. Are there any indications of how LLM would perform in problems with shallow plans? If the number of objects or actions is big enough, that might be challenging for classical planners.
>
> We considered this possibility too but found that LLMs don’t necessarily perform well in problems with shallow plans. We would like to point to the graphs in the pdf attached above as part of the global response. These graphs represent the distribution of the correct plans by GPT-4 over optimal plan lengths. From these graphs, we can say that our traditional notions of planning complexity do not hold with LLMs. For an LLM, an easier instance from the perspective of planning complexity is the same as a harder one as it just predicts the next tokens based on their weights and the context. We will update the Appendix with these discussions.
> > 2. Did you attempt relaxations with other planning problems? Perhaps it's not "natural" in blocks world, but there are other problems where the plans are equivalent to their relaxed versions.
>
> We have included the relaxation evaluation for other planning domains as well (Logistics and Mystery Blocksworld) in Appendix A.2.1.
> > 3. Did you try COT with PDDL prompts?
>
> We haven’t looked into chain of thought experiments with PDDL prompts, but we don’t expect the results to be any better.
> > 4. For the interactive scenario, does it make sense to keep the temperature at 0? Perhaps randomization might help the LLM to recover to deviate from earlier commitments.
>
> We had kept the temperature to be 0 primarily for the reproducibility of the results. We believe that it is an interesting additional investigation to play around with the temperature and check for performance improvement in the back-prompting method.
> > 5. What's I-GPT-3 in Table 3?
>
> I-GPT3 refers to the Instruct GPT3 model.
> > 6. The domain-specific translator is not discussed.
>
> For each domain, we perform template-based translation to translate from PDDL to natural language for the natural language prompt configurations. We will include this information in the paper as well.
> > 7. LLMs are trained in language and human-written code. The evaluation with Randomized Disguising might be less meaningful.
>     - However, Randomized Disguising is a small part of the work.
>     - An alternative not explored in the paper is to add human-readable description to the domains. Even though that requires human intervention, it's reasonable to assume the ones providing the domain can also provide a description.
>
> Our point in that domain obfuscation section was to show that LLM’s don’t seem to possess plan generation abilities that can’t be explained by their approximate retrieval abilities. To the extent LLM’s significantly worsen in plan generation when predicate names are changed either to other meaning-bearing words or random words, we believe it lends credence to our hypothesis.

---

> > ### Comment · Reviewer_NTC1 · 2023-08-14
> > **thank you**
> >
> > Thank you for your responses.
> > I agree that Randomized Disguising is not the same. That'd be a different experiment.

---

### Author Rebuttal · Authors · 2023-08-10

We thank the reviewers for their thoughtful feedback. We have provided our responses separately for each reviewer. We have attached a PDF containing the images and tables which we refer to in the individual responses.

---

### Decision · Program_Chairs · 2023-09-21

**Decision:**

Accept (spotlight)

**Comment:**

The paper investigates the ability of language models to plan on a set of commonsense planning tasks. It finds that though they are often unsuccessful at planning on their own, their use as a heuristic to guide search is more promising.

Overall the reviewers found the paper interesting, useful, and timely. They do note however the limited number of domains, which for the generality of LLMs would be quite useful to add. I agree with the reviewers (on both accounts) and recommend acceptance.

A few ablations such as CoT would improve the paper and new domains. The authors should consider including numbers for an open source model as well, though their point that GPT-4 is SoTA is well taken, but this would be useful for reproducibility (as GPT-4 is not a fixed, permanent checkpoint) and for those that build on the work.